# Predictive coding networks for temporal prediction

**Beren Millidge[1]◐, Mufeng Tang[1]◐, Mahyar Osanlouy[2], Nicol S. Harper[3], Rafal Bogacz[1]\***

**1** MRC Brain Network Dynamics Unit, University of Oxford, Oxford, United Kingdom, **2** Auckland Bioengineering Institute, University of Auckland, Auckland, New Zealand, **3** Department of Physiology, Anatomy and Genetics, University of Oxford, Oxford, United Kingdom

◐ These authors contributed equally to this work.
\* rafal.bogacz@bndu.ox.ac.uk

**Data Availability Statement:** All code required for replicating the simulation presented in this paper can be found freely online at https://github.com/C16Mftang/temporal-predictive-coding.

**Funding:** This work has been supported by the Biotechnology and Biological Sciences Research

## Abstract

One of the key problems the brain faces is inferring the state of the world from a sequence of dynamically changing stimuli, and it is not yet clear how the sensory system achieves this task. A well-established computational framework for describing perceptual processes in the brain is provided by the theory of predictive coding. Although the original proposals of predictive coding have discussed temporal prediction, later work developing this theory mostly focused on static stimuli, and key questions on neural implementation and computational properties of temporal predictive coding networks remain open. Here, we address these questions and present a formulation of the temporal predictive coding model that can be naturally implemented in recurrent networks, in which activity dynamics rely only on local inputs to the neurons, and learning only utilises local Hebbian plasticity. Additionally, we show that temporal predictive coding networks can approximate the performance of the Kalman filter in predicting behaviour of linear systems, and behave as a variant of a Kalman filter which does not track its own subjective posterior variance. Importantly, temporal predictive coding networks can achieve similar accuracy as the Kalman filter without performing complex mathematical operations, but just employing simple computations that can be implemented by biological networks. Moreover, when trained with natural dynamic inputs, we found that temporal predictive coding can produce Gabor-like, motion-sensitive receptive fields resembling those observed in real neurons in visual areas. In addition, we demonstrate how the model can be effectively generalized to nonlinear systems. Overall, models presented in this paper show how biologically plausible circuits can predict future stimuli and may guide research on understanding specific neural circuits in brain areas involved in temporal prediction.

## Author summary

While significant advances have been made in the neuroscience of how the brain processes static stimuli, the time dimension has often been relatively neglected. However, time is crucial since the stimuli perceived by our senses typically dynamically vary in time,

Council UK (https://www.ukri.org/councils/bbsrc/) grant BB/S006338/1 awarded to R.B., the Medical Research Council UK (https://www.ukri.org/councils/mrc/) grant MC_UU_00003/1 awarded to R.B., an E.P. Abraham Scholarship in the Chemical, Biological/Life and Medical Sciences awarded to M. T., and N.H was supported by Wellcome Trust (https://wellcome.org/) grant WT108369/Z/2015/Z. The funders had no role in study design, data collection and analysis, decision to publish, or preparation of the manuscript.

**Competing interests:** I have read the journal's policy and the authors of this manuscript have the following competing interests: BM and RB are shareholders in Fractile Ltd, which designs AI accelerator hardware.

and the cortex needs to make sense of these changing inputs. This paper describes a computational model of cortical networks processing temporal stimuli. This model is able to infer and track the state of the environment based on noisy inputs, and predict future sensory stimuli. By ensuring that these predictions match the incoming stimuli, the model is able to learn the structure and statistics of its temporal inputs and produces responses of neurons resembling those in the brain. The model may help in further understanding neural circuits in sensory cortical areas.

## Introduction

This paper is concerned with extending the theory of predictive coding to the problem of processing dynamically changing sequences of sensory inputs arriving over time. Predictive coding, which originated from an algorithm for compression in information theory [1], was initially developed and applied to the analysis of the brain by Srinivasa et al. [2] and Mumford [3] and then formalized into a general computational model of the cortex by Rao and Ballard [4]. The core hypothesis behind predictive coding is that the brain computes predictions of its observed input, and compares these predictions to the actually received input. The difference between the two is called the prediction error and quantifies how incorrect the brain's prediction was. Predictive coding proposes that the brain then adjusts its neural activities and synaptic strengths to minimize prediction errors which ultimately results in more accurate predictions [5, 6]. Thus, solely by minimizing prediction errors, the brain is forced to learn a general *world model* which can generate accurate predictions of its incoming sensory input [5, 7]. Moreover, these prediction errors are computed *locally* between the local input to the neuron and the predictions it receives. This means that learning in predictive coding model requires only local information and can be accomplished in most cases with purely Hebbian synaptic plasticity [8–10].

Predictive coding has become an influential theoretical model for understanding cortical functions [4, 5, 7, 11]. In their original study, Rao and Ballard trained predictive coding networks to generate static images of natural scenes, and demonstrated that the network learnt receptive fields with similarities to those found in V1, as well as reproduced extra-classical receptive fields and end-stopping [4]. Since then it has been demonstrated that predictive coding networks can explain many intriguing phenomena such as repetition suppression [12], bistable perception [13, 14], illusory motions [15], retinal stabilization [16] and even potentially attentional modulation of perception [17, 18]. Moreover, there has been much work matching the underlying neurophysiology of cortical microcircuits to the fundamental computations required by the predictive coding algorithm [19–21], thus providing a potential low-level basis for the implementation of predictive coding in neural circuitry.

Alongside the aforementioned works that have successfully reproduced many neurophysiological phenomena, recent progresses in predictive coding have been concerned with machine learning tasks such as the classification or generation of static images [22–25], and multiple lines of research have investigated the relationship between predictive coding and backpropagation, the driving force behind modern machine learning systems [22, 26]. However, most of these works are concerned with inputs that are independent and identically distributed (i.i.d.) samples from some datasets and are presented in batches to the predictive coding model in random order. However, the visual input to the brain is a continually changing sensory stream conveying a sequence of individual images with much correlation and rich structure embedded in the timing of the sequence elements. Therefore, to better describe the information

processing in the brain, predictive coding models must take into account one more crucial element: *time*.

There are several established algorithms in statistics and machine learning for sequence processing over time, but they typically require very complex computations that would be difficult for biological circuits to perform. Nevertheless, it is useful to consider them as a reference against which the performance of biologically plausible models can be assessed. When there is a linear relationship between the current and future states (and noise is assumed to be Gaussian), the optimal temporal predictions can be achieved by the Kalman filter [27]. For more complex nonlinear problems, one can employ recurrent neural networks [28], which contain recurrent connections which maintain and update an internal hidden state over time. While these recurrent networks, and more advanced successors such as the Long-Short-Term-Memory (LSTM) [29] can be very expressive and powerful, they are typically trained with backpropagation through time algorithm (BPTT) [30–32] which requires storing a history of all computations through a sequence and then 'unrolling' it sequentially backwards through time to make updates. This algorithm is biologically implausible, because the brain can only receive inputs in a sequential stream, and must be able to process them online, i.e. as the inputs are received, and seems unlikely to be able to unfold a precise sequence of computations in reverse. In this work, we focus our comparison to these statistical and machine learning models on the Kalman filter, an online algorithm that processes data sequentially, as biological systems appear to do.

Within the field of predictive coding, there are a few tracks of research that have considered incorporating the temporal dimension. Earlier works [33, 34] employed Kalman filtering to model the visual processing of dynamical sequences. Instead of using fixed transformation matrices which are assumed to be known as in Kalman filtering, these works introduced learning rules to model synaptic plasticity in neural circuits performing filtering. Friston et al. [35] have proposed the notion of extending predictive coding to use *generalized coordinates* [36, 37] which model a dynamical state by including a set of temporal derivatives in the state vector and making predictions of these derivatives along with the current state. A few recent studies have also developed predictive coding models that perform well in various tasks involving temporal dependencies, such as those commonly examined in machine learning [38–40]. However, the works mentioned above have departed from the simple and flexible architecture of classical predictive coding for static inputs [4] in order to take into account the temporal dimension. For example, the pioneering work that introduced Kalman filtering into predictive coding [33, 34] has not described how the computations of Kalman filtering could be implemented in biological networks. Mapping of models employing generalized coordinates [35] on a neural circuit would require explicit hard-coding of the expected temporal dependencies between different dynamical orders. Other models include specialised network features to aid the temporal processing, such as hyper-networks [40], multiple sub-networks [38] or complex connectivity and neuron types [39].

In this paper, we propose a simple predictive coding network that also incorporates the temporal dimension, which we call *temporal predictive coding* (tPC). This model generates predictions not only about the current inputs but also about its own *future* neural responses, which is achieved by recurrent connections between neurons to transmit the prediction of one time-step to the next. The paper makes four main contributions: First, we propose a predictive coding model that addresses the problem of temporal prediction, while inheriting from static predictive coding [4] the simple and biologically plausible neural network implementations employing only local connectivity and Hebbian plasticity rules. Second, when the model is linear, we show that our model is a close approximation of the Kalman filtering model analytically, and has empirical performance comparable to Kalman filtering in benchmark filtering

tasks while being computationally cheaper and requiring only biologically plausible operations. Also, unlike the Kalman filter, our model can learn the parameters of the system online. Third, when trained with natural moving stimuli, we find that the model develops localized, Gabor-like receptive fields similar to those observed by Rao and Ballard [4], and more importantly, the receptive fields are also motion-sensitive, a property unique to neurons responding to dynamic stimuli [41]. Finally, we extend the model to the nonlinear case and show promising performance on a number of nonlinear filtering and sequence prediction tasks. Overall, our model provides a possible computational mechanism underlying the cortical processing of dynamic inputs based on predictive coding, suggesting that the brain may learn to represent both static and continuous sensory observations using a single computational framework.

## Models

The structure of the exposition in this section is as follows. Firstly, we present the underlying graphical structure of our proposed generative tPC network i.e., a Hidden Markov Model (HMM). Next, we show that, with Gaussian assumptions on the HMM and certain parametric assumptions on the nonlinear generative processes, we can derive an objective function that is similar to the original predictive coding network [4] but taking into account the temporal dimension. We then demonstrate that neural dynamics and plasticity rules can be derived via the minimization of the objective function by gradient descent, and a corresponding training algorithm is presented. Finally, we show that the proposed computations and algorithms afford biologically plausible neural implementations in several different cases and discuss how they may be mapped to neural circuitry in the cortex. Since this paper contains much mathematical notation, for ease of reading, we have collected it in Table 1 for quick reference.

**Table 1. Table of mathematical notation used in the paper.** Vector and matrix variables are defined with their dimensions. Otherwise, the variables are scalars.

| Notation | Meaning |
|---|---|
| $y$ | Observation ($\mathbb{R}^{d_y}$) |
| $x$ | Latent state ($\mathbb{R}^{d_x}$) |
| $u$ | Control input ($\mathbb{R}^{d_u}$) |
| $k$ | Observation 'frame' |
| $t$ | Inference time |
| $A$ | Dynamics matrix ($\mathbb{R}^{d_x \times d_x}$) |
| $B$ | Control matrix ($\mathbb{R}^{d_x \times d_u}$) |
| $C$ | Observation matrix ($\mathbb{R}^{d_y \times d_x}$) |
| $\omega_x$ | Process noise ($\mathbb{R}^{d_x}$) |
| $\omega_y$ | Observation noise ($\mathbb{R}^{d_y}$) |
| $f$ | Nonlinear function |
| $\hat{x}$ | Inferred latent state ($\mathbb{R}^{d_x}$) |
| $q(\cdot)$ | Variational distribution |
| $\mathcal{F}$ | Variational free energy |
| $\epsilon_x$ | Dynamics prediction error ($\mathbb{R}^{d_x}$) |
| $\epsilon_y$ | Observation prediction error ($\mathbb{R}^{d_y}$) |
| $\eta$ | learning rate |
| $\Delta t$ | Inference step size |
| $\Sigma_k$ | Posterior variance ($\mathbb{R}^{d_x \times d_x}$) |
| $K$ | Kalman Gain matrix ($\mathbb{R}^{d_x \times d_y}$) |

## Model foundations: HMM and free energy

The conceptual level of our model is grounded within the Bayesian Brain paradigm [5, 42–44]. Specifically, we assume that the problem of perception is fundamentally an *inference* problem, where there exists some real world 'out there', from which we only receive noisy and distorted sensory input. We assume that the task of perception is to untangle and counteract the noise in order to reconstruct the real (but hidden) state of the world given only our sensory observations. Thus, mathematically, we can represent the problem of perception as trying to infer a series of latent states of the world $x_k$ ($k = 1, 2, \ldots$) from their corresponding sensory observations $y_k$ ($k = 1, 2, \ldots$). We assume that the underlying graphical structure of our tPC network is an HMM, where the hidden states $x_k$ follow a Markov chain. That is, the current hidden state of the world only depends upon the previous hidden state. Also, the current observation is generated by the current state of the world only, with no dependence on past noisy observations. Fig 1 shows the generative process of the tPC where, for generality, we have also added 'control' inputs $u$ which can be thought of as known inputs to the system at every time step (such inputs are included in the Kalman filter that we will later use as a benchmark). This control input is useful to model systems where there are known external forces acting on the system (such as an agent's own actions) which we do not necessarily want to model simply as part of the environmental dynamics.

From the generative model in Fig 1, we can also write out specific equations for the dynamics of the states and observations in what is called the state-space representation:

$$x_k = Af(x_{k-1}) + Bu_k + \omega_x, \tag{1}$$

$$y_k = Cf(x_k) + \omega_y \tag{2}$$

where $A$ is the matrix that transitions the previous hidden state $x_{k-1}$ to $x_k$, $B$ is the matrix governing how the control input $u_k$ affects the current hidden state, and $C$ is the 'emission' matrix that determines how the observation $y_k$ is generated from the hidden state. $f$ is a function transforming $x_k$ that may be nonlinear. The above state-space representation also includes sources of noise $\omega_x$ and $\omega_y$. We will assume a white Gaussian noise model such that $\omega_x \sim \mathcal{N}(0, \Sigma_x)$ and $\omega_y \sim \mathcal{N}(0, \Sigma_y)$ are zero-mean Gaussian random variables with covariance matrices $\Sigma_x$, $\Sigma_y$ (In the control theory literature, the process noise $\Sigma_x$ is often denoted as $Q$ and the observation

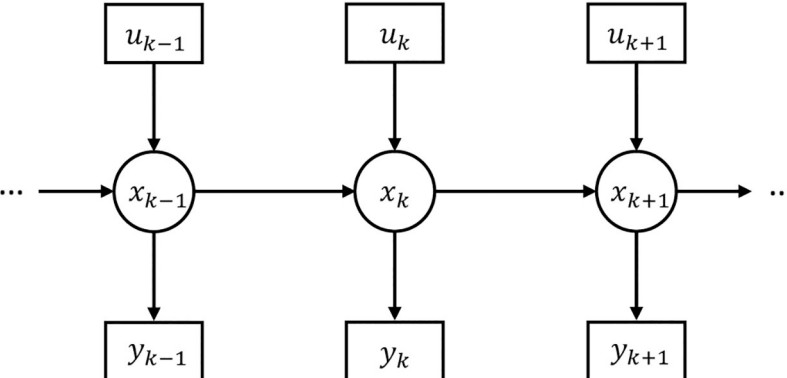

**Fig 1. Graphical model of the generative process assumed by temporal predictive coding.** $x_k$ correspond to hidden states, $y_k$ to observations, and $u_k$ to control inputs. Circles denote latent variables, squares denote observations, and arrows denote conditional dependence of the variables (the absence of an arrow indicates conditional independence).

noise $\Sigma_y$ as $R$. We instead follow the convention that has arisen within the predictive coding literature, which we also believe is more straightforward, by calling them $\Sigma_x$ and $\Sigma_y$, since it makes explicit that these variances are simply the variances of the two distributions composing the generative model.). Therefore, $x_k$ and $y_k$ can be considered as random variables that follow the Gaussian distributions:

$$x_k|x_{k-1}, u_k \sim \mathcal{N}(Af(x_{k-1}) + Bu_k, \Sigma_x),$$ 

(3)

$$y_k|x_k \sim \mathcal{N}(Cf(x_k), \Sigma_y).$$ 

(4)

The objective of our model is to obtain an estimate $\hat{x}_k$ of the current $x_k$, given the previously estimated $\hat{x}_{k-1}$, current noisy observation $y_k$ and control input $u_k$. This objective can be expressed as estimating the posterior distribution $p(x_k|y_k, \hat{x}_{k-1}, u_k)$. If function $f$ is nonlinear, an analytic expression for the posterior cannot be found, and thus variational inference [45–47] is utilized to find an approximate of the posterior. Specifically, we assume that there exists an approximate posterior $q(x_k)$ and we seek an approximate posterior that is as close as possible to the 'true' posterior $p(x_k|y_k, \hat{x}_{k-1}, u_k)$, and variational inference finds such a $q(x_k)$ by minimizing an upper bound on the divergence between the true and approximate posteriors called the variational free energy. If we make an additional simplifying assumption that $q(x_k)$ follows a Gaussian distribution with its density highly concentrated at its mean, the free energy $\mathcal{F}_k$ becomes approximately equal to [8, 9]:

$$\begin{aligned} \mathcal{F}_k &= -\log p(y_k, x_k|\hat{x}_{k-1}, u_k) \\ &= -\log p(y_k|x_k)p(x_k|\hat{x}_{k-1}, u_k) \end{aligned}$$ 

(5)

and it is also sufficient to estimate the mode of the approximate posterior (which is the same as its mean) instead of estimating the whole distribution with this assumption. Furthermore, with the Gaussian assumptions underlying Eqs 3 and 4, the free energy becomes (we choose to omit the other terms in the multivariate Gaussian density as they do not affect the optimization over $x_k$ and $A$, $B$, $C$):

$$\begin{aligned} \mathcal{F}_k &= \frac{1}{2}(y_k - Cf(x_k))^T \Sigma_y^{-1}(y_k - Cf(x_k)) \\ &\quad + \frac{1}{2}(x_k - Af(\hat{x}_{k-1}) - Bu_k)^T \Sigma_x^{-1}(x_k - Af(\hat{x}_{k-1}) - Bu_k). \end{aligned}$$ 

(6)

Importantly, we can express this objective as the sum of squared *prediction errors* weighted by their inverse covariances (which are often called precisions in the predictive coding literature). In this model, there are two kinds of prediction errors—'sensory' prediction errors which are the difference between the observation and predicted observation $y_k – Cf(x_k)$ and 'temporal' prediction errors which are the difference between the inferred current state and the current state predicted from the previous state $x_k - Af(\hat{x}_{k-1}) - Bu_k$. Thus, by finding an estimate $\hat{x}_k = \text{argmin}_{x_k} \mathcal{F}_k$, we effectively minimize the squared sum of these prediction errors while the precision matrices serve to weight the importance of the sensory and temporal prediction errors in accordance with their intrinsic variance (so highly variable prediction errors are weighted less). After the minimization finishes, the estimated $\hat{x}_k$ can then be used to estimate the hidden state at the next step $k + 1$.

### Inference and learning algorithm

With the objective function in Eq 6, it is then possible to derive an iterative algorithm to perform its minimization via gradient descent. Similar to static predictive coding [4], the gradient descent is performed on two sets of values: the hidden states $x_k$ and the weight parameter matrices $A$, $B$ and $C$. As we will show in the next subsection, the former can be implemented as neural responses and the latter can be implemented as synaptic connections in a neural circuit in a similar way to static predictive coding. For the hidden state $x_k$, its dynamics follow:

$$\tau \frac{dx_k}{dt} = -\frac{\partial \mathcal{F}_k}{\partial x_k}, \tag{7}$$

where $\tau$ is the time constant of the neurons. At convergence, we can say that the equilibrium value $\hat{x}_k$ represents the optimal inference about the mean of the true hidden state of the world. It is worth mentioning that we now introduced two different indices $k$ and $t$ for distinct time scales: for discrete steps $k$ at which the observations arise and continuous real time $t$ in which computations are made within the model, and we will discuss the relationship between them in detail in the next subsection.

To derive the exact expression for the inferential dynamics, we can define the precision-weighted state and observation prediction errors as $\epsilon_x$ and $\epsilon_y$ respectively as follows:

$$\epsilon_y = \Sigma_y^{-1}(y_k - Cf(x_k)), \tag{8}$$

$$\epsilon_x = \Sigma_x^{-1}(x_k - Af(\hat{x}_{k-1}) - Bu_k). \tag{9}$$

We can then write Eq 7 as:

$$\tau \frac{dx_k}{dt} = -\frac{\partial \mathcal{F}_k}{\partial x_k} = -\epsilon_x + f'(x_k) \odot C^T \epsilon_y, \tag{10}$$

where $\odot$ denotes the element-wise product between vectors. The dynamics have contributions from both the sensory and the temporal prediction errors, and the contribution of the sensory prediction error is 'mapped backwards' through the transpose matrix $C^T$.

Similarly, if we assume that the $A$, $B$ and $C$ parameter matrices are learnable, we can derive update rules also following gradient descent on $\mathcal{F}_k$:

$$\begin{aligned}
\Delta A &= -\eta \frac{\partial \mathcal{F}_k}{\partial A} = \eta \epsilon_x f(\hat{x}_{k-1})^T, \\
\Delta B &= -\eta \frac{\partial \mathcal{F}_k}{\partial B} = \eta \epsilon_x u_k^T, \\
\Delta C &= -\eta \frac{\partial \mathcal{F}_k}{\partial C} = \eta \epsilon_y f(x_k)^T.
\end{aligned} \tag{11}$$

In the above equations, $\eta$ denotes a scalar learning rate. As we will see below, the iterative updates will correspond to local Hebbian plasticity in the neural implementation of the model.

Typically, if the $A$, $B$, and $C$ matrices are learnt, then the matrices are updated by a single step according to Eq 11 after the $x_k$ has already converged and using the equilibrium values $\hat{x}_k$. This is because it is often assumed that these variables represent more slowly changing variables in the real world. Moreover, in the neural implementation, these matrices are often assumed to be implemented by synaptic strengths which typically change slowly while the $\hat{x}$ variables are typically mapped to neural firing rates, which change quickly.

The process of learning a tPC model is shown in Algorithm 1. The algorithm assumes that sensory observations $y_k$ arrive at discrete steps $k$. At each step $k$, we first initialize the $x_k$ values with the previous estimated $\hat{x}_{k-1}$ value from the last step. Then, we iterate Eq 10 until convergence for a given observation $y_k$. Upon convergence our $\hat{x}_k$ becomes our best estimate of the true state of the world. Given this $\hat{x}_k$, we can update the $A$, $B$, and $C$ matrices using Eq 11.

**Algorithm 1**: Single training epoch for temporal predictive coding

```
N: Discrete steps of observations
for k = 1 to N do
  Initialize xₖ with previously inferred x̂ₖ₋₁
  while xₖ not converged do
    Perform inference by updating xₖ (Eq 10)
  end
  Update weight matrices ΔA, ΔB, ΔC (Eq 11) using inferred x̂ₖ
end
```

## Neural circuit implementation

The update rules and dynamics we have derived in Eqs 10 and 11 can be mapped to a recurrent predictive coding network architecture with biologically plausible Hebbian learning rules. In this section, we present two examples of networks implementing the algorithm exactly, and then a simplified network that approximates the algorithm.

Fig 2A presents a potential example of how the update rules we have derived can be implemented in a neural network with only local and Hebbian plasticity, which is similar to the standard predictive coding network [4]. In this network, observations $y_k$ enter at the lowest level, and cause sensory prediction errors $\epsilon_y$ as the observations are met by top-down predictions. These prediction errors are explicitly represented by the activities of 'prediction error neurons'. These prediction error neurons receive top-down inhibitory connections from 'value neurons' in the layer above which, through their firing rates, represent the inferred posterior values $\hat{x}_k$. Similarly, at the layer above there are additional prediction error neurons that represent the difference between the current activity and that predicted based on the previous inference mapped through the dynamics function (Eq 9), and we assume that there are dedicated neurons that 'memorize' latent activities $\hat{x}_{k-1}$ inferred at the previous discrete time step, which is used to make a prediction of the current latent activities and reloaded at the end of inference at each time step.

There are several important aspects to note about this model. First, all the required update rules can be implemented using purely local information. The dynamics of the hidden state estimates $x_k$ (Eq 10) can be reproduced locally since the value neurons receive inhibitory connections from the prediction error neurons $\epsilon_x$ at their 'layer' as well as excitatory connections from the prediction error neurons $\epsilon_y$ at the layer below. Similarly, the prediction errors $\epsilon_y$ can be computed according to Eq 8 as the corresponding prediction error neurons receive excitatory input $y_k$ and inhibition from neurons encoding $x_k$. The prediction error $\epsilon_x$ can be computed analogously. The update rules for the $A$, $B$, and $C$ weight matrices (Eq 11) are also precisely Hebbian, since they are outer products between the prediction errors and the value neurons of the layer above which, crucially, are also precisely the pre and post-synaptic activities of the neurons where the synapses implementing these weight matrices are located. Moreover, we empirically demonstrate in the Results sections that scaling by the inverse covariance matrices $\Sigma_x^{-1}$ and $\Sigma_y^{-1}$ could be encoded in the learnt $A$ and $C$ matrices, similarly as it has been shown in static predictive coding models [48]. Thus, the tPC model can represent the covariance matrices implicitly in its synaptic weights, without needing to implement them in explicit

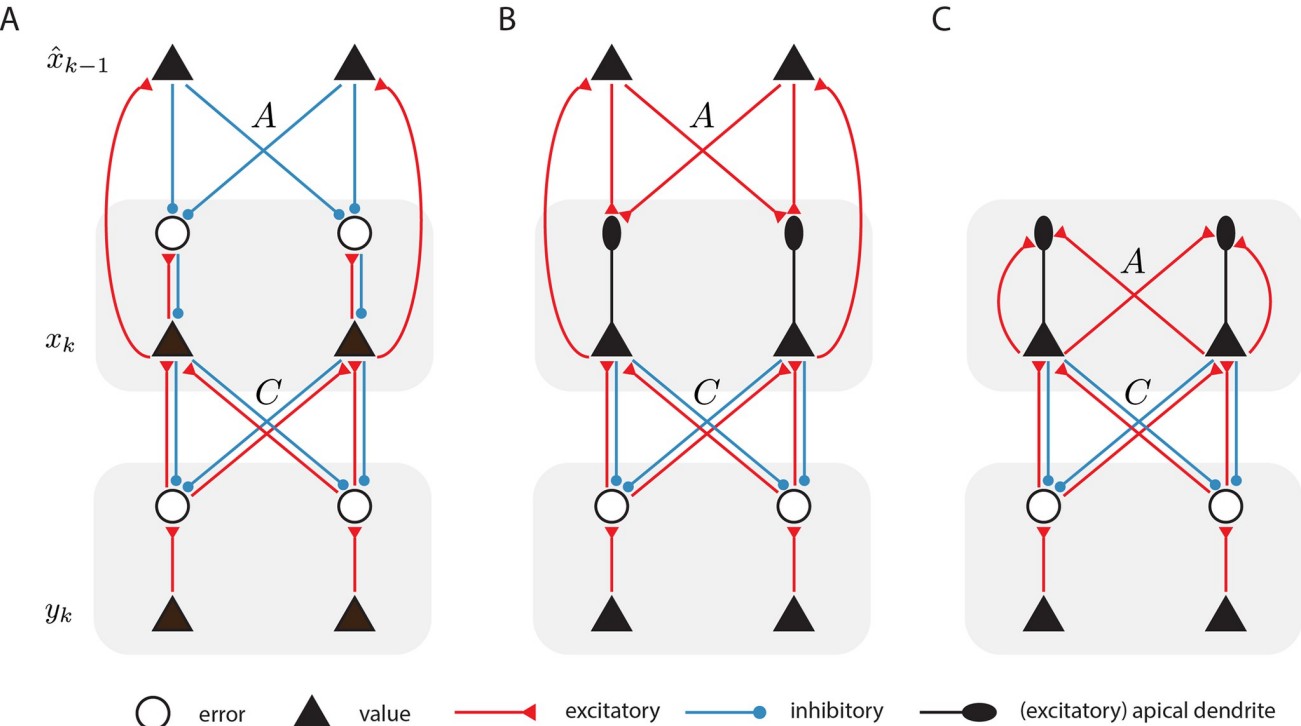

**Fig 2. Possible neural implementations of temporal predictive coding.** A: Potential neural circuit implementing the iterative recurrent predictive coding algorithm. For simplicity, we have depicted each neural 'layer' as possessing only two neurons. B: Version of the model where the prediction errors are represented by the difference in membrane potential in soma and at apical dendrites (depicted as ellipses). C: Neural circuitry required to implement the single-iteration predictive coding algorithms. This model no longer includes a separate set of neurons explicitly storing the estimate of the previous timestep, but instead, the temporal prediction errors are computed naturally through recurrent connections. For simplicity, we omitted the control inputs $Bu_k$, which can be implemented in a similar way to the recurrent inputs $A\hat{x}_{k-1}$ to the error neurons or apical dendrites.

synaptic weights. The nonlinear function $f(\cdot)$ can be implemented in this circuit following the way specified in [26], through local inhibitory neurons.

The network shown in Fig 2A follows a standard predictive coding architecture, but it could be simplified because the prediction error neurons encoding $\epsilon_x$ only project to corresponding neurons encoding $x$, and we could thus borrow the idea of dendritic computing similar to the model of [49]. In particular, substituting Eq 9 into Eq 10, we obtain:

$$\tau \frac{dx_k}{dt} = -x_k + Af(\hat{x}_{k-1}) + Bu_k + f'(x_k) \odot C^T \epsilon_y, \tag{12}$$

where, for clarity of explanation, we omitted the precision $\Sigma_x^{-1}$ that can be encoded in $A$. By writing the dynamical equation in this way, we assume that there is no building block within the model that encodes the error explicitly; rather, the apical dendrite will encode the inputs $Af(\hat{x}_{k-1}) + Bu_k$ and send the signal to the soma of the pyramidal neuron. This dendritic signal excites the soma and drives the inferential dynamics (Eq 12), together with the decay $-x_k$ that is intrinsic to the soma and feedback signals from the observation layer. The corresponding neural implementation is shown in Fig 2B. Although the architecture of the network becomes simpler, learning parameters $A$ and $B$ is less straightforward because the prediction error $\epsilon_x$ is not explicitly represented in the activity of any neurons in the network. Nevertheless, the prediction error $\epsilon_x$ is equal to the difference between the neural activity and the membrane

potential in the apical dendrite, and it has been proposed that such difference drives plasticity of synapses on the apical dendrite [50]. Since both the neural activity and the membrane potential are encoded within the same neuron, it is plausible that their difference could be computed within the neuron (e.g. the information on the neural activity could be brought to the synapses on apical dendrites via backpropagating action potentials). Such a signal could then drive local synaptic plasticity to learn the $A$ and $B$ matrices.

While simulating the model, we update the state estimates by numerically solving Eq 12 using the Euler method, i.e. we calculate the state estimates for every interval $\Delta t$:

$$x_k(t + \Delta t) = x_k(t) + \frac{\Delta t}{\tau}\left[-x_k(t) + Af(\hat{x}_{k-1}) + Bu_k + f'(x_k(t)) \odot C^T \epsilon_y\right]. \tag{13}$$

where $\frac{\Delta t}{\tau}$ is effectively the step size of the discretized inference that we tune in our simulations. The above expression highlights that the algorithm has two nested timescales—firstly there is the 'external' timescale which is where sensory inputs $y_k$ are received in a sequence of steps we index by subscript $k$. Then, for each external step, there is an internal inference of the hidden state that is numerically implemented as a set of recurrent iterations within that external step, which we denote as $t$. In such a nested framework, the implementations in Fig 2A and 2B need to store and hold fixed the state estimate of the previous step while the iterative inference of the state estimate for the current step is ongoing. Specifically, the estimate from the previous step $\hat{x}_{k-1}$ needs to be held fixed throughout the iterative procedure while the actual current values $x_k(t)$ vary in order to find a balance $\hat{x}_k$ between the demands of matching the prediction from the last step and also the current observation (Algorithm 1). Once the iterations are complete for a step, the new value of $\hat{x}_k$ needs to be loaded into the memory and stored as the last step for the next set of iterations. In situations where the observations are separated in time, it is known that neurons are able to store the representation of stimuli presented a few seconds earlier in their activity [51]. In the case where sensory input arrives continuously, the two timescales coincide, and there may be no time for inference between steps of sensory input. In that latter case, the algorithm can be adapted to remove the issue of nested timescales without unduly harming filtering performance by simply using a *single* iteration of the internal inference for each external step. This means that there is effectively no 'inner loop' of the algorithm anymore, since the inner loop consists of just a single iteration. This makes the algorithm fully online in the sense that it receives a new sensory input for every step. In particular, note that by equating time indices $\Delta k = \Delta t$, we obtain:

$$\hat{x}_k = x_k + \frac{\Delta t}{\tau}\left[-x_k + Af(\hat{x}_{k-1}) + Bu_k + f'(x_k) \odot C^T \epsilon_y\right]. \tag{14}$$

A diagram of a potential neural circuit implementing this single-step algorithm is presented in Fig 2C. This network no longer includes neurons storing past inferences. Instead, the temporal prediction errors are computed solely using recurrent connections labelled $A$, which are now assumed to introduce a temporal delay of one step.

The advantage of this approach is that it eschews the challenge of storing and loading the memory of the last step; instead, this memory can be dynamically maintained across a single external step simply through recurrent connections via their intrinsic synaptic delays. The disadvantage of this approach is that the update rules of the algorithm were derived as gradient descent on an objective function, and this approach is equivalent to taking only a single gradient step for each example. Clearly, in many cases, such an algorithm simply would not work because a single step is nowhere near enough to approach the optimum. However, there are two features of the problem that ameliorate much of this difficulty in practice. The first is that, when $f$ is a linear function, the objective is actually convex, and thus the loss landscape is

extremely well-behaved. This allows for the use of relatively high integration steps to move large distances in a single, or a few steps, without fear of overshooting the optimum or running into divergences. The other factor is due to the nature of the external world: typically, visual scenes change relatively slowly on a microsecond-by-microsecond level, and thus the optimal estimated hidden state in a single step is likely close to the optimum hidden state for the next. In this case, since we initialize the inference of each step with the optimum of the last step, this will usually be close to the optimum for the current step as well, thus meaning that the algorithm simply does not have to make many iterations to achieve the optimum since it already starts close by. In the next section, we show that, in practice, on standard tracking and filtering tasks, these two factors can often simplify the inference problem enough that this single iteration approach often works successfully, although it usually does not perform quite as well as multi-step methods.

## Results

The results of this paper are partitioned into a theoretical section and experimental sections. In the theoretical section, we examine the relationship between the tPC model and Kalman filtering and demonstrate that the tPC network, under certain assumptions, is equivalent to a Kalman filter with a fixed posterior variance. In simulations, we demonstrate that, despite not correctly representing the posterior variance, the tPC network nevertheless exhibits strong and robust tracking performance on both linear and nonlinear filtering tasks while also being capable of online system identification through the learning of the $A$ and $C$ matrices, unlike the Kalman filter. Moreover, we show that when tPC is trained with natural movies, the simulated neurons in the latent layer develop motion-sensitive receptive fields resembling those of neurons in the primary visual cortex.

### Relationship to Kalman filtering

Here we show that both Kalman filtering [27] and tPC can be derived as special cases of the *Bayesian filtering* problem. Bayesian filtering concerns the problem of inferring the *sequence* of hidden 'causes' $x_1, \ldots, x_K$ of the observations $y_1, \ldots, y_K$. This problem can be effectively factorised into a sequence of online inference problems i.e., inferring the hidden state $x_k$ at time step $k$, given the whole history of observations $y_1, \ldots, y_k$ [52, 53]. For simplicity of notation, we denote $x_{1:k} = x_1, \ldots, x_k$ and $y_{1:k} = y_1, \ldots, y_k$. The Bayesian filtering problem can thus be formulated as inferring the following posterior distribution:

$$p(x_k|y_{1:k}). \tag{15}$$

We show in S1 Appendix that this posterior distribution is proportional to:

$$p(x_k|y_{1:k}) \propto p(y_k|x_k) \int p(x_k|x_{k-1})p(x_{k-1}|y_{1:k-1})dx_{k-1} \tag{16}$$

where the integral is effectively the marginal distribution $p(x_k|y_{1:k-1})$ of the joint $p(x_k, x_{k-1}|y_{1:k-1})$ and can be considered as the prior on $x_k$. Notice that the term $p(x_{k-1}|y_{1:k-1})$ is exactly the posterior inferred from the previous time step $k-1$, making Bayesian filtering a recursive method [54]. As a special case of Bayesian filtering, Kalman filtering assumes that the conditional distributions $p(y_k|x_k)$ and $p(x_k|x_{k-1})$ can be parameterized *linearly* as follows:

$$y_k|x_k \sim \mathcal{N}(Cx_k, \Sigma_y); \quad x_k|x_{k-1} \sim \mathcal{N}(Ax_{k-1} + Bu_k, \Sigma_x). \tag{17}$$

Further, it assumes that the posterior estimated at the previous step $k - 1$ follows:

$$x_{k-1}|y_{1:k-1} \sim \mathcal{N}(\hat{x}_{k-1}, \Sigma_{k-1}) \tag{18}$$

where $\hat{x}_{k-1}$ is the MAP estimate from the previous step $k - 1$ and is the mode (or mean) of the Gaussian posterior with covariance $\Sigma_{k-1}$ at step $k - 1$. Under the above Gaussian assumptions, the prior $p(x_k|y_{1:k-1})$ on $x_k$ (i.e., the integral in Eq 16) can be written as [55]:

$$p(x_k|y_{1:k-1}) = \mathcal{N}(A\hat{x}_{k-1} + Bu_k, A\Sigma_{k-1}A^T + \Sigma_x). \tag{19}$$

Kalman filtering then performs maximum a posteriori (MAP) to find $\hat{x}_k$:

$$
\begin{aligned}
\hat{x}_k &= \underset{x_k}{\operatorname{argmax}} \log p(x_k|y_{1:k}) \\
&= \underset{x_k}{\operatorname{argmin}} \, (y_k - Cx_k)^T \Sigma_y^{-1}(y_k - Cx_k) \\
&\quad + (x_k - A\hat{x}_{k-1} - Bu_k)^T(A\Sigma_{k-1}A^T + \Sigma_x)^{-1}(x_k - A\hat{x}_{k-1} - Bu_k).
\end{aligned}
\tag{20}
$$

Since all the transformation functions in this optimization problem are linear, an analytical expression for $\hat{x}_k$ can be derived, which will result in the well-known algorithm for Kalman filtering i.e., the 'projection':

$$
\begin{aligned}
\hat{x}_k^- &= A\hat{x}_{k-1} + Bu_k \\
\Sigma_k^- &= A\Sigma_{k-1}A^T + \Sigma_x
\end{aligned}
\tag{21}
$$

and the 'correction' step where we then incorporate the new information we have received from the environment to correct our estimates:

$$
\begin{aligned}
\hat{x}_k &= \hat{x}_k^- + K(y_k - C\hat{x}_k^-) \\
\Sigma_k &= (I - KC)\Sigma_k^- \\
K &= \Sigma_k^- C^T[C\Sigma_k^- C^T + \Sigma_y]^{-1}
\end{aligned}
\tag{22}
$$

where $K$ is known as the Kalman Gain matrix and is central to the Kalman filter update rules for the estimated mean and variance in the correction step [56]. $\hat{x}_k$ and $\Sigma_k$ are then our estimated mean and covariance of the posterior Gaussian distribution. The derivation of the projection and correction rules can be found in prior works [54, 57], while we also provide the derivation in S2 Appendix as to compare to the update rules of our tPC model, which is demonstrated below.

Our tPC model also aims to solve the Bayesian filtering problem in Eq 16, and it can also make the linear and Gaussian assumptions underlying Eq 17. Notice that Eq 17 is identical to Eqs 3 and 4, but with a linear $f$. tPC differs from Kalman filtering by making a different assumption on the distribution on the previous-step posterior $p(x_{k-1}|y_{1:k-1})$. Instead of assuming it as a Gaussian distribution in Eq 18, it assumes:

$$x_{k-1}|y_{1:k-1} \sim \delta(x_{k-1} - \hat{x}_{k-1}) \tag{23}$$

where $\delta(x_{k-1} - \hat{x}_{k-1})$ denotes a Dirac distribution with its density concentrated at $\hat{x}_{k-1}$. The prior on $x_k$ for predictive coding thus becomes:

$$p(x_k|y_{1:k-1}) = p(x_k|\hat{x}_{k-1}) = \mathcal{N}(A\hat{x}_{k-1} + Bu_k, \Sigma_x) \tag{24}$$

since the density of $p(x_{k-1}|y_{1:k-1})$ is concentrated at $\hat{x}_{k-1}$. The MAP estimation of $\hat{x}_k$ performed

by predictive coding is thus:

$$
\begin{aligned}
\hat{x}_k &= \underset{x_k}{\arg\max} \log p(x_k|y_{1:k}) \\
&= \underset{x_k}{\arg\min} \, (y_k - Cx_k)^T \Sigma_y^{-1} (y_k - Cx_k) \\
&\quad + (x_k - A\hat{x}_{k-1} - Bu_k)^T \Sigma_x^{-1} (x_k - A\hat{x}_{k-1} - Bu_k)
\end{aligned}
\tag{25}
$$

which is the free energy in Eq 5 with a linear *f*. The above derivation thus provides another interpretation of the tPC model, i.e. a special case of the Bayesian filtering problem that assumes at each step the variance estimated at the previous step is 0. Again, as all transformations in the optimization objective are linear, we can derive an analytical expression for $\hat{x}_k$:

$$
\begin{aligned}
\hat{x}_k^- &= A\hat{x}_{k-1} + Bu_k \\
\hat{x}_k &= \hat{x}_k^- + K(y_k - C\hat{x}_k^-) \\
K &= \Sigma_x C^T [C\Sigma_x C^T + \Sigma_y]^{-1}
\end{aligned}
\tag{26}
$$

which is similar to the projection and correction steps for Kalman filtering (Eqs 21 and 22), but with $\Sigma_{k-1}$ assumed to be 0, i.e., tPC does not propagate the uncertainty estimations. The derivation of these equations can be found in S2 Appendix.

It is also worth mentioning that in the tPC model, although we did not specify the estimation of uncertainty $\Sigma_k$ at each step, not *propagating* the uncertainty $\Sigma_{k-1}$ from the previous step is not equivalent to not *estimating* it. In fact, as was shown in [58], even when we choose to estimate $\Sigma_k$ in the tPC model, it is still not propagated. Therefore, our MAP estimation $\hat{x}_k$ will not be affected.

To summarize, by assuming a linear *f*, here we have shown that both Kalman filtering and tPC are special cases of the Bayesian filtering problem, while the key difference is that predictive coding ignores the uncertainty estimated at the previous step when estimating the most likely hidden 'cause' at the current time step, whereas Kalman filtering always estimates this uncertainty. However, as we will show in the experimental results section, in the benchmark tracking tasks, tPC performs on par with Kalman filtering, albeit not estimating the posterior uncertainty. Importantly, there are several advantages of tPC over Kalman filtering as a model of dynamical processing in the brain. Firstly, the projection and correction steps of Kalman filtering require complicated matrix algebra and are challenging to compute in neural circuitry, especially the Kalman Gain matrix *K*. On the other hand, although we can derive analytical results for the estimates of tPC as well, these estimates can be obtained via the iterative methods mentioned above, which afford plausible circuitry implementations (Fig 2). Secondly, the iterative nature of our tPC model also makes it adaptable to nonlinear *f*, where there are no analytical solutions to the Bayesian filtering problem. In contrast, extending the Kalman filter to nonlinear systems is challenging and standard methods such as the extended Kalman filter [59] work by linearizing around the nonlinearity and thus require knowledge of the Jacobian of the nonlinearity at every state, which is also challenging to implement in neural circuits. Finally, the Kalman filter assumes knowledge of the correct *A*, *B*, and *C* matrices while these must presumably be learnt from sensory observations in the brain.

## Performance in linear filtering problems

Here we first present results for the linear tPC model on a simple tracking task of the kind to which the Kalman filter is commonly applied in industry. Here, the goal is simply to infer the unknown hidden state (position, velocity, acceleration) of an object that is undergoing an

unknown acceleration. We receive noisy observations of the position, state, and acceleration of the object which are mapped through a random $C$ matrix with additional observation noise. We use a random $C$ matrix for the observation mapping to simulate and test the most difficult scenario where the observations are entirely scrambled.

Mathematically, the generative process of this task can be represented according to a linear state space model. We assume that the position and velocity of the object follow the usual laws of Newtonian physics, and there is a persistent acceleration which is affected by the control input, giving us the following $A$ and $B$ matrices,

$$
A = \begin{bmatrix} 1 & \Delta k & \frac{1}{2}\Delta k^2 \\ 0 & 1 & \Delta k \\ 0 & 0 & 1 \end{bmatrix}, \ B = \begin{bmatrix} 0 & 0 & 1 \end{bmatrix}. \tag{27}
$$

In Eq 27, $\Delta k$ denotes the duration of the interval between successive sensory observations (we used $\Delta k = 0.001$). Additionally, we draw a fixed $C$ matrix from a random Gaussian distribution $C \sim \mathcal{N}(0, 1)$, and modeled the control inputs as $u_k = e^{-0.01k}$. The process and observation noise $\Sigma_x$ and $\Sigma_y$ were set to identity matrices. We then generate the true latent states $x_k$ and the noisy observations $y_k$ using Eqs 1 and 2, initialized with a zero vector when $k = 0$. The performance of the models is then measured as the mean squared error (MSE) between the estimated $\hat{x}_k$ and true $x_k$ across all observation time steps. Fig 3A shows an example of the true system state that we generated across 1000 time steps, and Fig 3B shows its corresponding noisy observations. As can be seen, the projected observations are completely scrambled by matrix $C$, making it a challenging task for the models to retrieve the true system states.

We then investigate tracking performance using tPC compared to the Kalman filter for both 5 steps of inference between observations ($\Delta k = 5\Delta t$) and 1 step ($\Delta k = \Delta t$). Since the problem is linear, we also investigate the performance of a tPC when its inference dynamics have reached the equilibrium, using the equilibrium condition derived in Eq 26. Since tracking performance is visually indistinguishable when zoomed out over 1000 timesteps, in Fig 3C, we plot the estimates of acceleration ($x_3$) on the 40 timesteps between 560 and 600 steps in. It can be seen from Fig 3C that both the tPC model with 5 inference steps and that with fully converged inferential dynamics could achieve comparable performance to the Kalman filter. Interestingly, the estimates of tPC tend to be closer to those of the Kalman filtering, rather than the true values. Although the tPC model with a single inference step (corresponding to the neural implementation in Fig 2C) has worse tracking performance, it is able to estimate a smoothed version of the trajectory of the system state. We hypothesize that the smoothed estimate with a single inference step is likely due to the fact that the tPC model does not completely converge in 1 iteration, and so does not completely optimize its estimate on every timestep, with the effect that the estimate is less sensitive to new information and effectively averages over recent experiences rather than optimally solving each one independently. A similar performance comparison is obtained on the position ($x_1$) and velocity ($x_2$) and is shown in S1 Fig.

To quantify the effect of the number of inference iterations and integration step size $\frac{\Delta t}{\tau}$ upon performance, in Fig 3D we plotted the MSE difference between predictive coding models with various inference iterations and inference step sizes and the Kalman filter, which is the optimal solution to the tracking problem. The MSE is calculated as the mean squared difference between the estimated system state $\hat{x}_k$ and true state $x_k$, averaged across time steps and trials. We find that with a small number of inference steps, the performance of the tPC model is worse, indicating that additional steps of inference aid the tracking performance of the

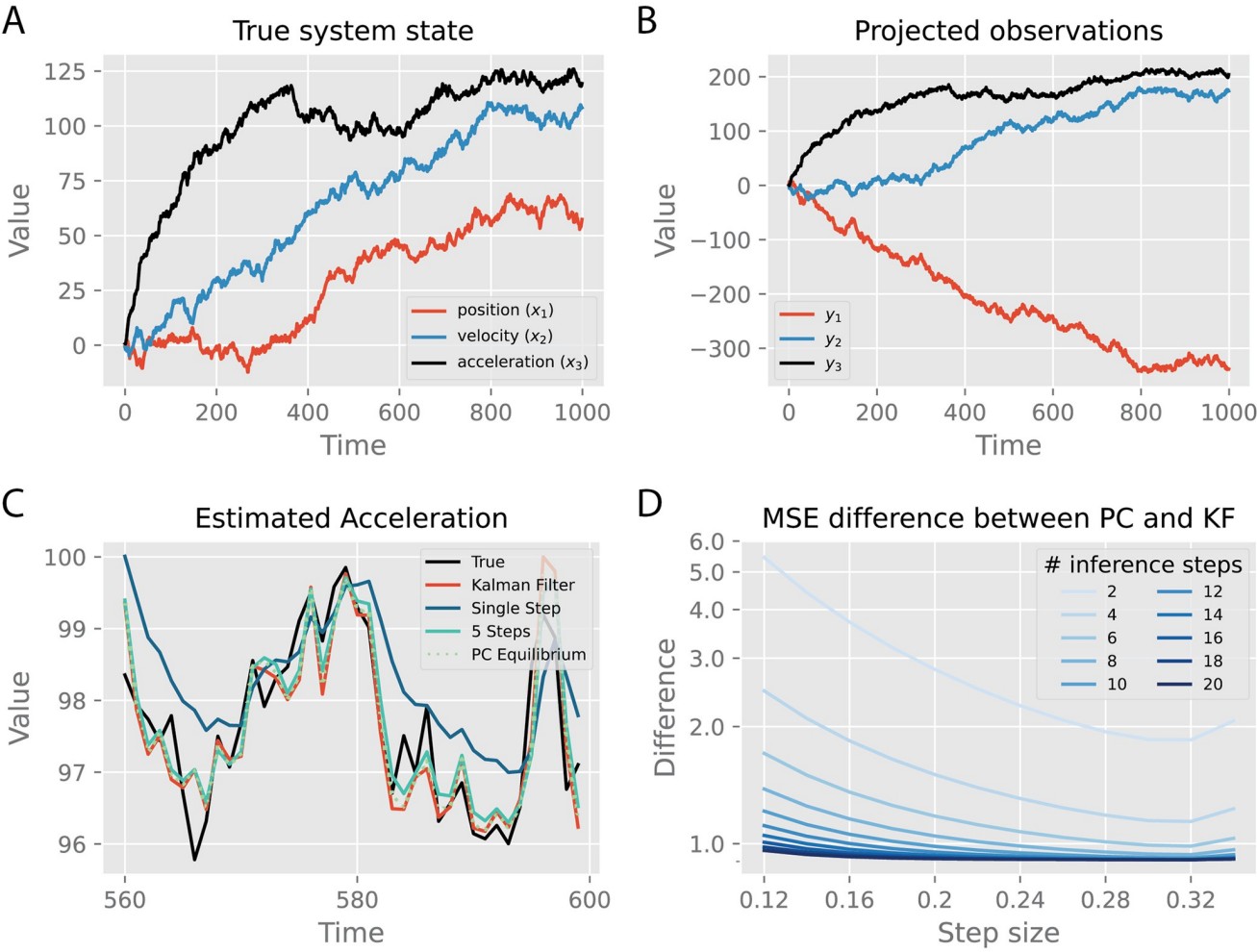

**Fig 3. The tracking task and the impact of inference step size and the number of inference steps on performance.** A. The dynamics of the true hidden state are represented as a 3-dimensional vector at each time step, with entries corresponding to position ($x_1$), velocity ($x_2$) and acceleration ($x_3$). B. The projected noisy observations from the true system state in A. C: Estimates of the acceleration with different models, zoomed in at the interval between 560 and 600 time steps. D: MSE difference between tPC and Kalman filter, with varying numbers of inference steps and step sizes for predictive coding. PC stands for temporal predictive coding and KF stands for Kalman filter. All values are with arbitrary units (a.u.).

algorithm. Moreover, although increasing the step size will initially improve the performance, the MSE will start to increase if the step size is too large. It can also be seen that with more inference steps and appropriate step sizes, the performance of tPC will be able to approximate that of the (optimal) Kalman filter.

## Learning the synaptic weights

In the previous investigations, we fixed the parameters $A$, $B$ and $C$ to the true values and only performed the inference dynamics. However, in many cases, simply inferring the hidden states of the world is not enough because we cannot assume that we know a-priori the structure of the dynamics or observation functions of the world. That is, in most real-world situations, the $A$, $B$, and $C$ matrices are unknown. Instead, we must *learn* these matrices from observations. In the tPC model, we have a natural Hebbian plasticity-based learning rule which we can use

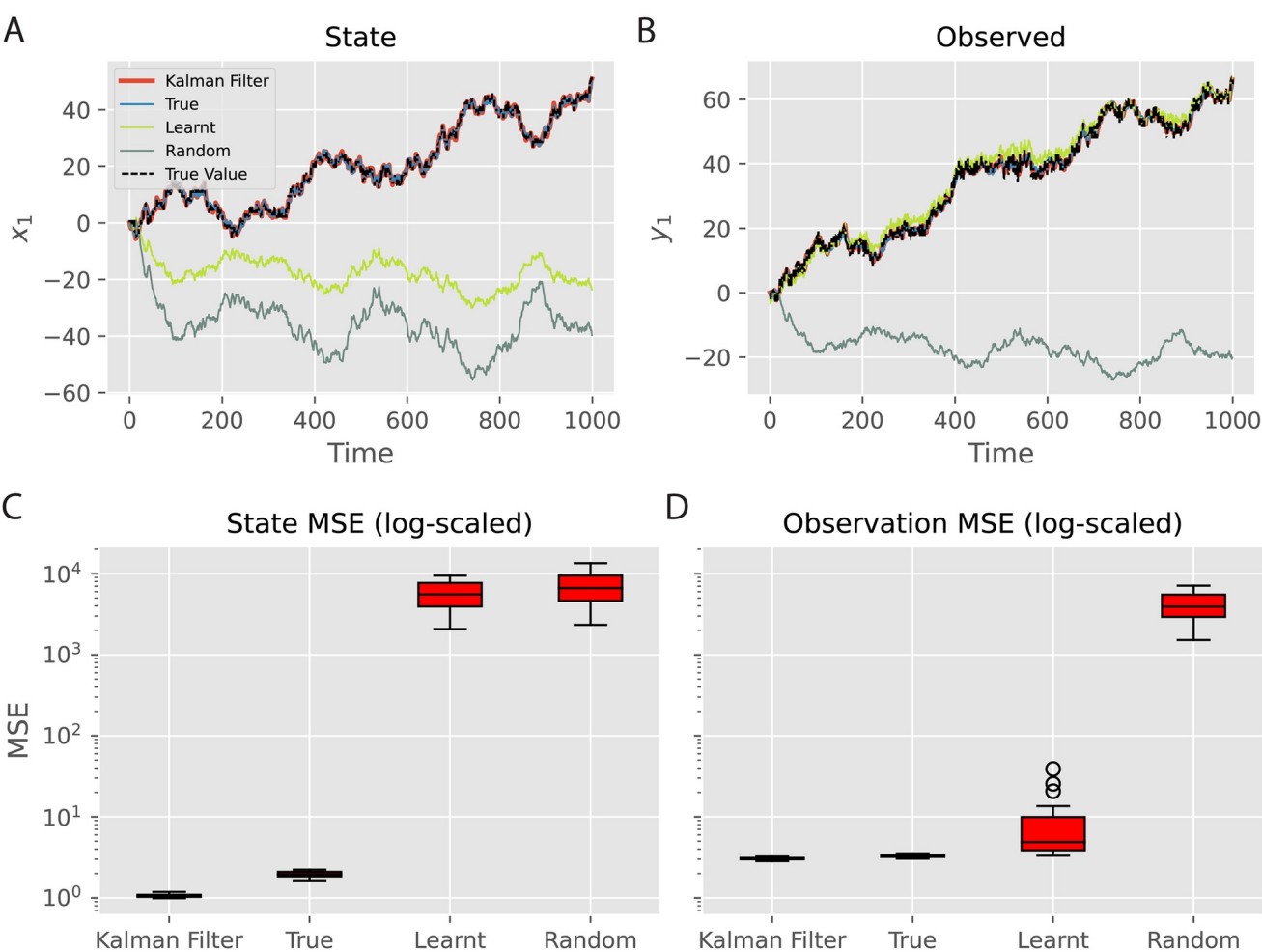

**Fig 4. Effects of learning parameters *A* and *C*.** A, B: Estimation of the state and observation trajectories respectively by different models. 'True', 'Learnt' and 'Random' denote the predictive coding model with true, learnt and random *A* and *C* respectively. Only the first dimension of the latent and observation is shown for simplicity. The other two dimensions have similar performance. C, D: MSE of the predictions on the hidden and observation levels respectively. Boxplots were obtained with 40 trials for each model. Both *x* and *y* are with arbitrary units (a.u.).

to learn these matrices directly (Eq 11). Here, we investigate how learning these parameters affects the performance of the tPC model. Specifically, we compare three different ways of setting the values for *A* and *C*: 1) fixing them to the true values used for generating the data; 2) learning them using Eq 11 and Algorithm 1; 3) fixing them to random values. We then examine the performance of these models on two levels, the latent state level (*x*) and the observation level (*y*), by measuring how well the model estimates the activities on both levels. It is worth noting that the observation estimates are calculated by performing a forward pass at each time step i.e., $\hat{y}_k = C(A\hat{x}_{k-1} + Bu_k)$, where the values of *A* and *C* are obtained at each time step *k* via the aforementioned three approaches. The results are shown in Fig 4.

For this set of results, we use 20 inference steps with a step size 0.2 to get the optimal performance for the tPC models, based on Fig 3D. Fig 4A shows that, while our tPC estimates the latent state well with true *A* and *C*, when asked to learn the parameters, the model fails to accurately estimate the latent state. Quantitatively, as shown in Fig 4C, the estimation MSE of the learning model on the state level is similar to tPC with totally random parameters, which is

much higher than those of the Kalman filter and tPC with true parameters. On the other hand, however, we find that the model learning $A$ and $C$ can accurately estimate the observations even with the incorrectly estimated latent state (Fig 4B). This effect arises because the problem of inferring the true hidden state from the data is fundamentally under-determined. There are many possible hidden states that, given a flexible learnt mapping, could result in an identical predicted observation. Importantly, despite inferring a different representation of the hidden state, the network is able to learn $A$ and $C$ that correctly predicts the incoming observations (Fig 4D).

## Learning the noise covariance matrices

For all our experiments above, we have used $\Sigma_x = \Sigma_y = I$ when generating the training data, where $I$ is the identity matrix. However, the noise covariance underlying a natural dynamic process may not always be identity. Although earlier works have proposed to encode the noise precision matrices $\Sigma_x^{-1}$ and $\Sigma_y^{-1}$ into additional connections explicitly [9], this approach would introduce extra complexity into the neural implementation of tPC. On the other hand, it has been shown that in the static case, the noise precision matrix can be implicitly encoded in recurrent connections similar to the $A$ matrix in our tPC model [48], without needing to represent the precision matrix explicitly as in [9]. Therefore, here we investigate whether $A$ and $C$ can encode the precision matrices of the process and observation noise respectively after learning. We used the same $A$, $B$ and $C$ matrices for data generation as before, but set the noise covariance matrices as:

$$\Sigma_x = \Sigma_y = \begin{bmatrix} 10 & 0 & 0 \\ 0 & 1 & 0 \\ 0 & 0 & 1 \end{bmatrix} \text{ or } \Sigma_x = \Sigma_y = \begin{bmatrix} 10 & 2 & 0.5 \\ 2 & 1 & 0.4 \\ 0.5 & 0.4 & 1 \end{bmatrix}. \tag{28}$$

We refer to the first case as 'non-identity diagonal' and the second case as 'positive definite'. The choice of these covariance matrices is arbitrary, although we intentionally make one diagonal entry larger than the others to examine tPC's capability of learning such a unique structure. The training hyper-parameters for this experiment are identical to the ones used for Fig 4, with which the learning for $A$ and $C$ could also converge. As Fig 5A shows, when the noise covariance matrices are non-identity diagonal and positive definite, the recurrent matrix $A$ is able to encode the large diagonal entry 10 in $\Sigma_x$ into its diagonal entries. Likewise, the $C$ matrix develops stronger weights to account for the larger variance of the inputs, suggesting that our tPC model can learn the noise covariance $\Sigma_x$ and $\Sigma_y$ into its recurrent and feedforward weights, without accounting for them in the energy function and learning/inference dynamics explicitly.

We then conducted a quantitative analysis of the impact of non-identity noise covariance in Fig 5B and 5C, similar to Fig 4. Here, both 'Kalman Filter' and 'True' use the correct $A$, $C$, $\Sigma_x$ and $\Sigma_y$, although the 'True' tPC model performs inference to get the hidden states using Eqs 8, 9 and 10. The 'Learnt' model has no access to all the matrices and has to learn both the generative process and the noise covariance with its weights. The results are similar to Fig 4: the model that learns $A$ and $C$ will fail to learn the correct hidden states but its observation estimates are on par with Kalman filtering, even when the noise covariance matrices are non-identity. Interestingly, we also observed slightly degraded latent estimation performance of the 'True' model with non-identity noise covariance. We hypothesize that this is due to the large

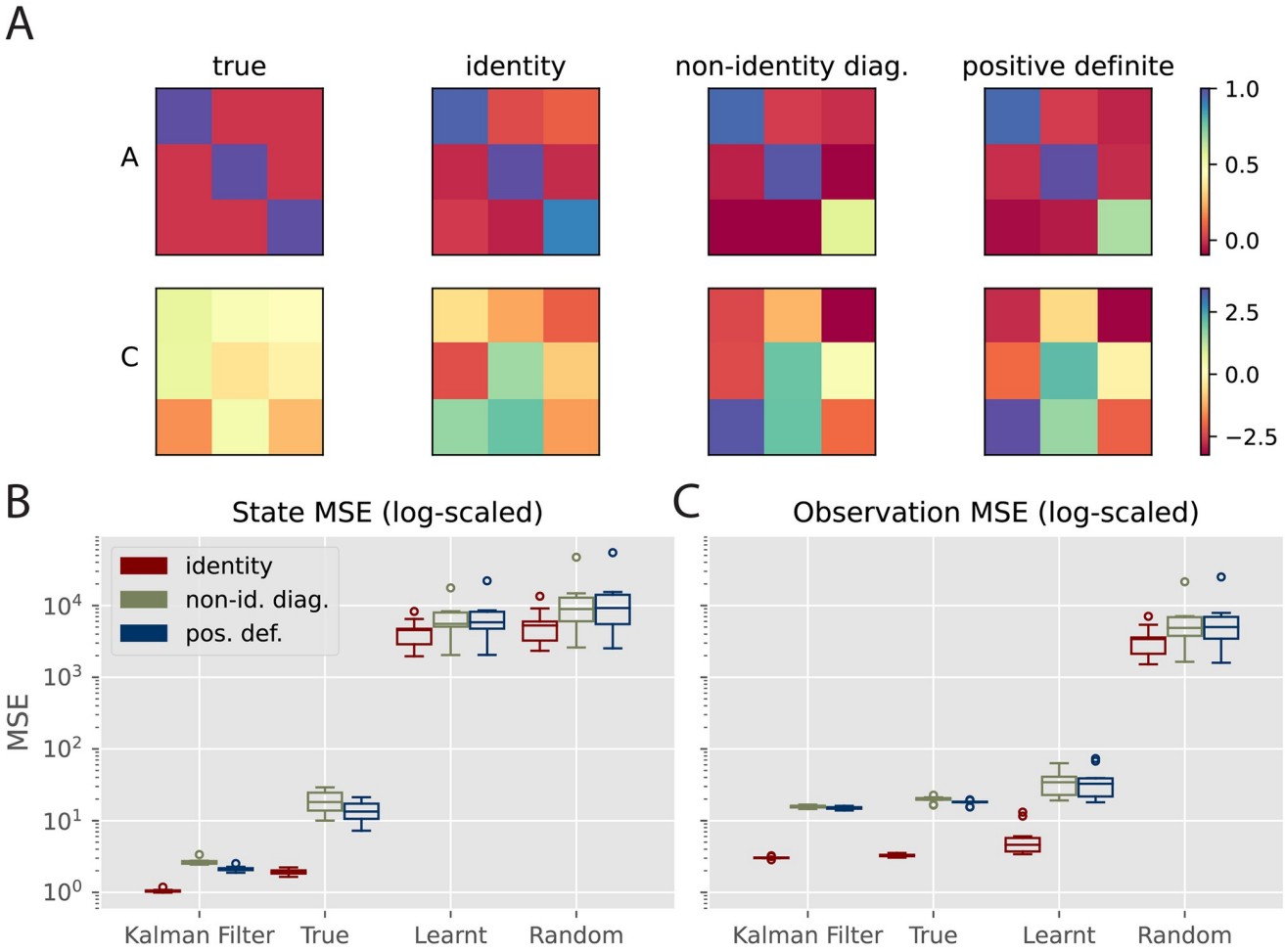

**Fig 5. Performance with non-identity noise covariance.** A: True and learnt *A* and *C* matrices with different underlying noise covariance matrices. B, C: MSE of the predictions on the hidden and observation levels with different noise covariance matrices. Error bars obtained with 40 trials.

diagonal entry 10 affecting the inference dynamics, which can be solved using coordinate-wise inference step sizes.

Overall, these results suggest that the synaptic plasticity of tPC can encode the noise covariance in the generative process, without representing them explicitly in the dynamics and circuit implementations. It is also interesting to investigate the exact theoretical relationship between the weights and noise covariance, similar to the analytical relationship in [48], and we intend to investigate it in future explorations.

## Training tPC with natural movies

Thus far, we have examined tPC in low-dimensional examples to understand its computational properties. In this section, we demonstrate that this model can also be applied to high-dimensional stimuli and provide a plausible account of how the biological visual system develops representations of dynamical inputs. To do so, we trained the tPC model on a dataset of patches extracted from movies of natural scenes. Each of the movies consisted of 50 frames of $200 \times 200$ pixels that we spatially bandpass filtered by a retina-like centre-surround filter [60].

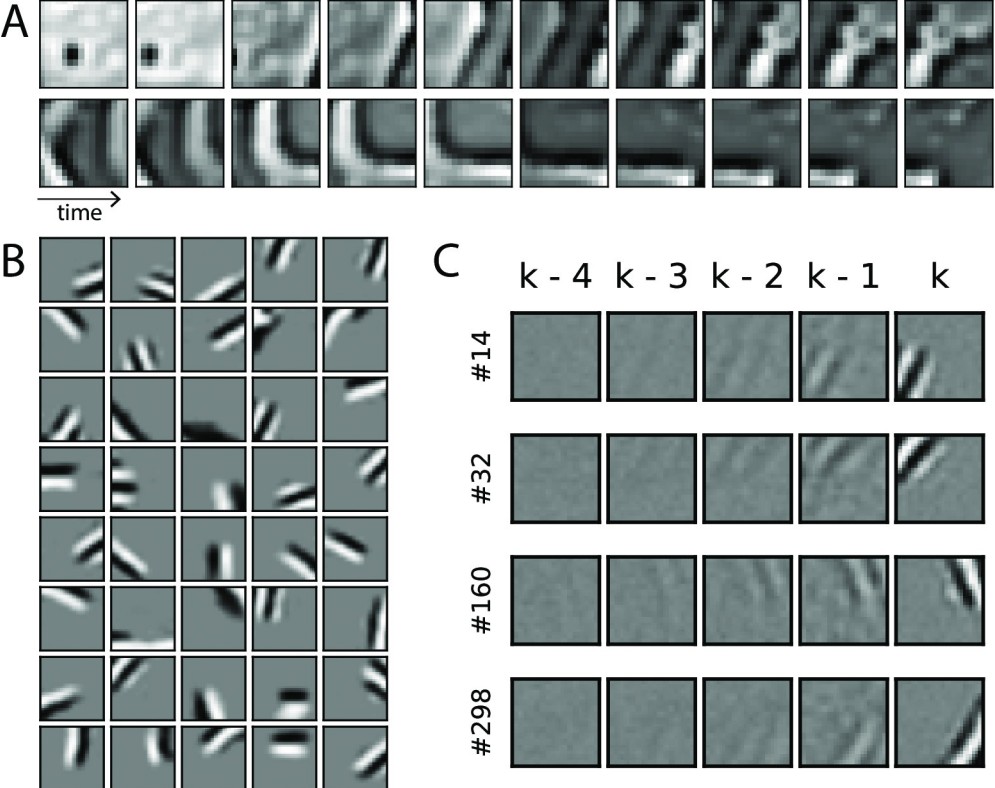

**Fig 6. Representations developed by the model when trained with patches from movies of dynamic natural scenes.**
A: First 10 frames of 2 example training movies used in our experiments. Patches extracted from movies obtained at websites pexels.com, pixabay.com and commons.wikimedia.org (for wikimedia attributions see https://github.com/C16Mftang/temporal-predictive-coding). B: The projective fields $C$ developed Gabor-like filters after training. C: Space-time receptive fields developed by hidden neurons of the tPC model.

We extracted $16 \times 16$ patches from the movies with the highest motion to form a dataset of 2000 moving patches of 50 frames. We also augmented the dataset with a left-right flipped version of itself to ensure a rich variety of motion directions. Two examples of the first 10 frames of the movies can be seen in Fig 6A. We then trained a tPC model with 320 hidden neurons with this dataset. Specifically, due to the redundancy of information in natural scenes and neural connectivity and energy constraints, we trained a tPC model with $L1$ sparsity-constraint latent activities and forward weights $C$. Therefore, the objective function we used in this set of experiments is:

$$\mathcal{F}_k = \|y_k - Cf(x_k)\|_2^2 + \|x_k - Af(\hat{x}_{k-1})\|_2^2 + \lambda_x |x_k|_1 + \lambda_C \sum_{i,j} |C_{ij}| \qquad (29)$$

where $|\cdot|_1$ denotes vector 1-norm, and we used a linear $f(\cdot)$ in these experiments. We also assume there is no control input to the model, so $B = 0$. The sparsity constraints are similar to the classical sparse coding model [60] and recent temporal prediction neural network models of dynamical inputs [61, 62]. To describe the cortical processing of natural scenes more accurately, for the experiments with these natural movies, we also used a time constant $\tau = 10ms$ estimated for visual cortical neurons [63]. The natural movies used in this experiment were recorded with frame rates in the range of 20Hz to 30Hz. In our simulations, we set $\Delta t$ to $0.1ms$ (and therefore the inference step size $\frac{\Delta t}{\tau} = 0.01$) and perform inference for 330 iterations to

(approximately) match the 30Hz frame rate, although we did not observe any qualitative difference when we used more inference iterations to match the lower frame rates.

We then trained our adjusted model with the natural movies. In Fig 6B we show the (sparsity-constraint) forward weight $C$ (of size $256 \times 320$), which develops Gabor-like and localized projective fields similar to those observed in mammalian primary visual cortex [64]. This finding is unsurprising as such a sparsity-constraint generative model is similar to the original predictive coding [4] and sparse coding models [60], which also reproduced localized Gabor filters.

To study the dynamical properties of tPC, we then performed a reverse correlation analysis [65] on the hidden neurons of the trained tPC. Specifically, after training, we supply a sequence of white noise stimuli to the model, let the hidden neurons relax according to Eq 13 to develop latent representations of the stimuli, and average the stimuli giving more weights to those producing the highest response of a neuron. In particular, for each time step $k$ and each neuron, we multiply the hidden activity $x_k$ with each of the white noise stimuli from $y_{k-4}$ to $y_k$ and sum up the products. The weighted sum is the spatio-temporal receptive field (STRF) of this neuron [65]. Formally, the STRF of the $i$th hidden neuron is defined as:

$$\text{STRF}_i = \frac{1}{T-5} \sum_{k=5}^{T} x_{i,k} y_{k-4:k}, \tag{30}$$

where $x_{i,k}$ denotes and activity of the hidden neuron $i$ at time step $k$ and $y_{k-4:k}$ denotes the 5 frames of white noise preceding and including time step $k$. Some examples of the STRFs developed by the hidden neurons of the tPC are shown in Fig 6C, where each row denotes a neuron and each column a time step. Two important properties emerge: 1) The STRFs are temporally asymmetric i.e., the power of the receptive fields decays back in time. This is consistent with observations in real neurons [66] and earlier computational models [61, 62]; 2) Importantly, many neurons also develop motion-sensitive STRFs, i.e., the spatial regions each neuron is responsive to shift location in the visual field over history steps, moving in a fixed direction (right-ward for neurons 160 and 298, left-ward for neurons 14 and 32 in the shown examples). Such motion sensitivity has been observed in real neurons in early visual areas [41, 67] and demonstrated in earlier sparse coding models [68, 69]. Overall, these results demonstrate that the tPC model with realistic time constants matching cortical neurons can reproduce neural representations observed in the visual areas, providing a possible computational mechanism underlying the learning of such representations.

## Extending tPC to nonlinear tasks

Here we examine our tPC model in nonlinear tasks. In order to make the task more challenging, we train the model on a simulated motion of a pendulum, in which the generative process does not explicitly follow that of tPC. Fig 7A shows a free-body diagram of the pendulum that we simulated in this experiment, demonstrating the mass, length $L$, and force vectors acting on the system. We describe the state of the pendulum by the angle of the pendulum $\theta_1$ and its angular velocity $\theta_2$. According to Newton's Second Law of Motion, the angle of the pendulum evolves according to:

$$\ddot{\theta}_1 = -\frac{g}{L} \sin(\theta_1) \tag{31}$$

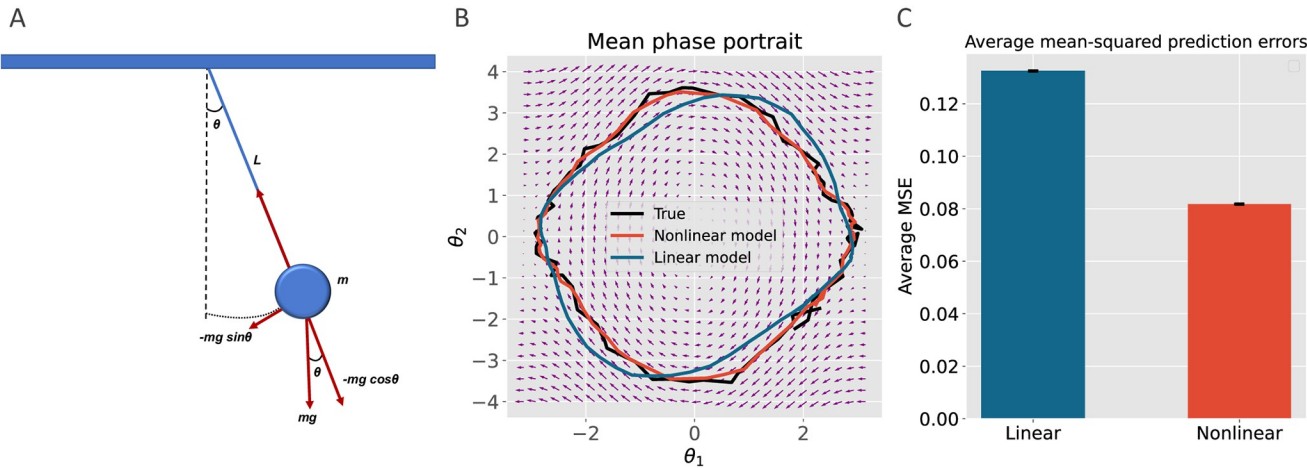

**Fig 7. Simulations of the pendulum.** A: A free-body diagram of a simple pendulum that has a mass $m$ attached to a string with length $L$. Also shown are the forces applied to the mass. The restoring force $-mg \sin \theta$ is a net force toward the equilibrium position. B: A phase portrait of the pendulum simulation showing the result of our linear versus nonlinear models prediction for the ground-truth data. The vector field (i.e. set of small arrows) was created by computing the derivatives of $\frac{d\theta_1}{dt}$ and $\frac{d\theta_2}{dt}$ at $t = 0$ on a grid of 30 points over the range of $-\pi$ to $+\pi$ and -4 to +4 for $\theta_1$ and $\theta_2$, respectively. C: The barplot shows the difference between the mean prediction errors of the linear model versus the nonlinear model from 100 simulations with varying noise profiles. The mean errors are significantly different ($p \ll 0.001$).

where $g$ is gravity. We can express this motion in a set of first-order equations:

$$
\begin{aligned}
\dot{\theta}_1 &= \theta_2 \\
\dot{\theta}_2 &= -\frac{g}{L} \sin(\theta_1).
\end{aligned}
\tag{32}
$$

In the simulation, we used the following parameters: $g = 9.81$ m/s$^2$, $L = 3.0$ m. We then simulated the system as an initial value problem by numerically integrating the equations using the explicit Runge-Kutta method for 2500 seconds with $\Delta k = 0.1$ second time steps and initial values of $\theta_1 = 1.8$ rad and $\theta_2 = 2.2$ rad/s. These values were chosen to simulate the pendulum motion with a large amplitude of oscillation to shift the system into a more nonlinear regime. The time series presented to the models $y_k$ were created by adding these numerical solutions and Gaussian noise with a zero mean and standard deviation of 0.1.

We trained both nonlinear (with a hyperbolic tangent nonlinearity $f(\cdot)$) and linear tPC models to predict these time series. Parameters of $C$ and $A$ were initialized to the identity matrix and the zero matrix, respectively. The learning was performed in the same fashion as in the previous section.

Fig 7B shows the results from the pendulum simulations using the phase portrait of the system. The solutions of the ground-truth simulation and our nonlinear model prediction are plotted on the vector field for the final 80 seconds. Even though both models performed relatively well by correctly predicting the behaviour of the pendulum motion, the linear model performed worse when the pendulum reached its highest angular displacements (see the noisy prediction around the extremities of the $\theta_1$ axis in Fig 7B). On the other hand, the nonlinear model stayed relatively stable throughout the entire prediction. Fig 7C shows the mean squared error of prediction averaged over 100 separate simulations, and the nonlinear model consistently outperformed the linear one. In S3 Appendix we also examined a nonlinear task

where the data generation process follows the generative model in Eqs 1 and 2 and obtained similar results.

## Discussion

In this paper, we have analysed the recurrent predictive coding architecture for temporal prediction and filtering. This task is important because processing time-varying sequences of inputs and using them to infer dynamically changing hidden states of the world is often considered a core task of the sensory regions of the brain [70]. As such, it is likely that these regions have an architecture heavily specialised for performing such filtering tasks. Here we have shown that the filtering problem can be tackled with a simple and biologically plausible algorithm with a straightforward implementation in neural circuitry.

We have derived our tPC model from first principles as a variational filtering algorithm, providing a clear algorithmic derivation in terms of gradient descents on the resulting free energy functionals. We have also proposed a direct implementation in neural circuitry which relies on only local information transmission as well as purely Hebbian plasticity that inherits the simple neural implementation of static predictive coding networks. Furthermore, we have also demonstrated that in the linear case, the algorithm is closely related to Kalman filtering, and is capable of robustly solving filtering and tracking tasks at a level close to the optimal linear solution. Moreover, and unlike the Kalman filter, we have demonstrated that our model can perform online *learning* of the parameters using Hebbian plasticity, which works rapidly and effectively in predicting the correct observations. Importantly, when trained on natural stimuli and constrained by sparse weights and activities, the tPC model develops motion-sensitive Gabor-filters of the visual scene, which is consistent with representations developed in the visual cortex. We have also extended the algorithm to the nonlinear case. We demonstrated that, when presented with noisy nonlinear stimuli, the nonlinear model has a superior performance over a linear model in both learning the dynamics and predicting the behaviour of the input sensory observation.

### Related work

Several earlier works have tried to approach the problem of Kalman filtering in the brain. Wilson and Finkel [71] repurpose a line attractor network and show that it recapitulates the dynamics of a Kalman filter in the regime of low prediction error. However their model only works for a single-dimensional stimulus, does not encode uncertainty, and also only works when a linearisation around zero prediction error holds. Deneve et al. encoded Kalman filter dynamics in a recurrent attractor network [72]. Their approach however encodes stimuli by means of basis functions, which leads to an exponentially growing number of basis functions required to tile the space as the dimensionality of the input grows. In the predictive coding approach, neurons directly encode the mean of the estimated posterior distribution, which means that the network size scales linearly with the number of dimensions. Our gradient method also completely eschews the direct computation of the Kalman Gain, which simplifies the required computations significantly. Additionally, Beck et al. show that probabilistic population coding approaches can compute posteriors for the exponential family of distributions of which the Gaussian distribution is a member [73]. However, no explicitly worked-out application of the population coding approach to Kalman filtering exists, to our knowledge. Recent work has also addressed the question of how biological recurrent neural networks can be trained for temporal prediction [74]. They proposed that synapses maintain eligibility traces encoding to what extent they contributed to neural activity over time, and when combined

with error signals, such traces enable effective credit assignment. It would be interesting to investigate how such eligibility traces could be incorporated into tPC networks.

Moreover, other works have also explored predictive coding for temporal predictions. For instance, early works [33, 34, 75] utilized a Kalman filter combined with sparse image representations to make future predictions of visual stimuli. However, these works did not describe how the Kalman filter can be implemented in biological circuits, and how their Kalman filter-based models can be extended to the nonlinear case. More recently, Jiang and Rao [40] trained a temporal version of predictive coding on sequences of natural video (filmed by a person walking through a forest) and observed that neurons have spatiotemporal receptive fields resembling those in the primary visual cortex. However, unlike our model, their model relies on an external hyper-network to perform temporal predictions. Temporal versions of predictive coding networks have also been extended to include multiple levels of hierarchy [39, 40]. Analysis of these networks revealed that neurons on higher levels change their activity with a slower time scale than the neurons at the lower levels of the hierarchy [40]. It has been also demonstrated that hierarchical temporal predictive coding networks can achieve performance comparable to BPTT in standard machine learning benchmarks [39]. However, these models require complex neural network implementations to perform these temporal tasks. It is thus an interesting future direction to see whether our tPC model, which inherits the simple neural implementation of the static predictive coding network, can present similar performance and neural responses.

Work by Lotter et al. [76] adapted deep recurrent neural networks to perform a kind of predictive coding whereby the network was trained to predict future *prediction errors* of each layer. They demonstrated that the resulting network was capable of correctly predicting sequences of video frames. While substantially scaling up predictive coding architectures to challenging machine learning tasks, the networks of Lotter et al. [76] diverged in many ways from classical predictive coding architectures, and also utilized many non-biologically plausible components from machine learning such as convolutional and LSTM layers as well as training their network with BPTT.

Kutschireiter et al. [38] addressed the question of how temporal predictive coding networks can be extended so they represent posterior uncertainty. They demonstrated that if multiple copies of the network are made, and the dynamics of each network include noise with an appropriate magnitude, then each network can represent a sample from the posterior distribution $p(x_k|y_1...y_k)$ and the collection of networks as a whole can represent the posterior distribution of state in a sampling-based manner. Their model is particularly interesting because the posterior uncertainty can be decoded from the differences in the activity of individual networks. However, the encoding of posterior uncertainty comes with the cost of a larger number of neurons required to form multiple networks.

A number of studies have used normative models to generate spatiotemporal receptive fields resembling those of direction-selective V1 simple cells [61, 62, 68, 69, 77–82]. Some models involved mechanisms related to predictive coding, such as sparse coding [68, 69] and independent component analysis [77] and applied them to spatiotemporal stimuli. However, while the resulting receptive fields were sometimes direction-selective, they did not have the asymmetric temporal profile seen in real STRFs. Application of slowness principles to spatiotemporal stimuli can also produce direction-selective STRFs [80, 81], but they again lack the asymmetric temporal profile. Other models have trained single-hidden-layer neural networks to perform temporal prediction on movies of natural scenes [61, 62, 82]. This has been shown to reproduce direction-selective STRFs with an appropriate asymmetric temporal profile [61], and indeed when stacked hierarchically reproduces units resembling motion-sensitive simple, complex and pattern-motion cells [62]. However, these models were trained by back-

propagation, arguably a biologically-unrealistic learning mechanism, hence while they are informative about whether the cortex might have temporal prediction as a normative objective they are agnostic about the potential learning mechanism. Our model helps bridge this gap, demonstrating how temporal prediction can be combined with a more plausible learning mechanism to produce units with spectrotemporal receptive fields resembling those of direction-selective V1 simple cells.

### Relationship to Kalman filtering

The similarities and differences between the tPC algorithm and classical filtering algorithms like Kalman filtering are of significant theoretical interest, as earlier works by Rao and Ballard [34] have already used Kalman filtering as a model of dynamical processing in the brain, and recent works have also been interested in Kalman filtering in a biologically plausible setting [83]. We have found that the crucial distinction is in the representation of the model's uncertainty, mathematically represented in the posterior variance $\Sigma_k$. Specifically, in the tPC model, although it represents the two 'objective' uncertainties $\Sigma_x$ and $\Sigma_y$ of the dynamical system it is inferring, it does not represent the uncertainty in the estimated posterior distribution (unlike the extended model of [38]). Crucially, it is the assumption of the prior at each time step that prevents the tPC model from successfully propagating the posterior uncertainty through time.

Although tPC does not optimally update and represent the dynamically changing posterior uncertainty of the agent, it has some computational advantages over Kalman filtering, which may render it more suitable for implementation in neural circuitry. The key advantage of the predictive coding approach is its computational simplicity compared to the Kalman filtering update rules which require complex matrix algebra and especially matrix inversions to compute the Kalman Gain matrix which are unlikely to be implementable in neural circuitry directly, while the predictive coding equations are simple and only require local and Hebbian updates and can be directly translated into relatively straightforward neural circuits. Moreover, as seen in Fig 3, the predictive coding estimate and the Kalman filter estimates of a given dynamical system end up closely converging anyway, which means that tPC networks could provide the brain an efficient and cheap way to approximate the highly effective Kalman filter using only simple circuitry.

One reason why predictive coding networks achieve performance similar to the Kalman filter is that the posterior uncertainty decays rapidly over trials (for an illustration see Fig 5A in Moeller et al. [84]). Furthermore, for deterministic transition processes (with $\Sigma_x = 0$), the posterior uncertainty decays to $\Sigma_k = 0$ [84]. Therefore, as the learning progresses, the Kalman filter becomes more similar or even identical (for deterministic processes) to tPC.

Our work thus raises an interesting empirical question as to what kinds of uncertainties various agents—such as humans or animals—actually appear to represent in dynamical inference tasks. For instance, it is not clear in the literature that subjective confidence ratings are highly correlated with the true dynamical uncertainty of the decisions in a task [85]. Although the Kalman filter has been used to describe reinforcement learning [86], direct comparison with simpler reinforcement learning models did not favour the Kalman filter [87]. It may be interesting, therefore, to compare predictions of the Kalman filter (and the extended model representing posterior uncertainty [38]) and the original tPC models to experimental data directly. For example, one could compare if the learning rate in reinforcement learning tasks is better described by the Kalman Gain or the value from tPC. Another interesting line related to Kalman filtering is to compare nonlinear tPC with extended [88] and unscented [89] Kalman filters in nonlinear tasks. Since this work focuses on linear tPC and its properties, we leave the exploration of these directions for future investigations.

## Relationship with generalized coordinates

A further interesting theoretical property of the model is its potential to autonomously learn to represent the dynamics of systems in generalized coordinates of motion [35, 36] if provided with generalized coordinates as inputs. Generalized coordinates, introduced into the predictive coding literature by Friston et al. [36], and well known in engineering practice, involve representing the n'th order derivatives of a state as additional coordinate dimensions. In effect, a point in generalized coordinates of motion to n'th order reflects the approximate trajectory of the state given by the n'th order Taylor expansion around that point. Original predictive coding models involving generalized coordinates typically required hardcoding the relationships between the coordinates, and the relative precisions between different dynamical orders [35, 90, 91] which results in intricate and complex hardcoded connectivity, reducing the ultimate biological plausibility of such models. However, our model's capability to directly learn the $A$ and $C$ matrices from data allows the model to simply receive a generalized coordinate state as input and learn the required connectivity online, as demonstrated in our pendulum simulations.

## Neural implementation

There are several interesting questions regarding the biological plausibility of the multi-step neural implementation. Initially these schemes, while they arise directly from the gradient descent derivation, appear biologically implausible for two reasons. The first is the issue of storage. Iterative schemes require the initial conditions (state estimate $\hat{x}_{k-1}$) to be held fixed throughout multiple iterations, and this means that this state must be stored somewhere accessible to be utilized multiple times during the iterative inference phase. It is not clear where or how this information could be stored in the brain, especially in low-level sensory systems. This storage must be local and ubiquitous as a naive implementation of the multi-step algorithms proposed in this paper would require separate storage for every single value neuron.

The second issue relates to the time it takes for an iterative algorithm to converge. Specifically, if we model the brain as receiving visual input as a continuous stream, then multiple iterations based upon a single stimulus would necessitate ignoring the data arriving in the intervals while the iterations are taking place. Moreover, an iterative approach would also take more time to update upon information newly received, which could be crucial for survival in some cases. There are also multiple potential solutions to this problem—firstly, the cortex may implement both an iterative and an amortized feedforward pass solution simultaneously [92], and there is evidence for precisely this. Firstly, core visual functions can often occur within 100–200 ms [93–96] which is too short to allow multiple steps of recurrent optimization, thus demonstrating that some kind of rapid single-step inference is possible. Conversely, there is much evidence that increased viewing time allows for the refinement of representations, reduction of uncertainty, and improvement in accuracy over time, which strongly speaks to the existence of some iterative recurrent processing occurring as well.

Finally, there is some interesting evidence that most brain regions, including the visual cortex, operate on a characteristic frequency [97, 98]. In the case of the visual cortex, the dominant rhythm is the alpha band at 5–15 Hz. Experiments have found that information presented in phase with these oscillations is processed normally; however, if the information is presented out of phase, then a drop of accuracy ensues, suggesting that the information has not been fully or successfully processed [99]. These findings are consistent with the iterative convergence algorithms proposed here being implemented in the cortex.

A further avenue for future work relates to the challenge of learning long-term dependencies which span over many time steps. This has long been a central challenge with these

recurrent models, and emerges essentially due to the fact that information is permuted or lost at every step of the recurrent pass, and thus tracking dependencies across many recurrent loops becomes increasingly difficult [29, 100, 101]. Numerous solutions to this have been suggested in the literature, ranging from specially designed cell architectures that can explicitly store or pass along information unaltered [29, 101] to having a nested hierarchy of recurrent models which allow for the propagation of information over longer and longer timescales [102].

The recurrent tPC model we propose can also be implemented when representing prediction errors in dendrites as in [49] and [103] instead of using explicit prediction error neurons as in Fig 2A. Such an architecture can reconcile predictive coding networks with the lack of strong evidence for there being explicit prediction error neurons in the cortex [11], unlike the dopaminergic reward prediction error neurons in the mid-brain whose existence has been established for decades [104, 105]. For example, a recent study in rat cortex found little evidence for prediction-related signals in spikes, but found strong evidence for it in local field potentials, which are thought to be driven by somatic and dendritic potentials [106]. Although in Fig 2B and 2C we still use error neurons in the hierarchical part of the network, they can also be circumvented following the dendritic implementation in [107]. Moreover, as was discussed in [19], it is also possible to implement tPC with differentiation between excitatory and inhibitory neurons following Dale's Law, further increasing the biological plausibility of the model.

## Conclusion

In this paper, we have proposed temporal predictive coding, a model of how the processing of dynamic stimuli may take place in the brain. We have demonstrated that the tPC model performs simple and localised computations that afford a biologically plausible neural network implementation while approximating the Kalman filtering model in theory. We have empirically verified this approximation, and further demonstrated that tPC can reproduce motion-sensitive receptive fields in the visual areas of the brain. Moreover, the model can be straightforwardly extended to account for nonlinear dynamics, bringing an insight on how the brain may perform dynamic sensory processing that is often nonlinear in nature.

## Supporting information

**S1 Appendix. Derivation of recursive Bayesian estimation.**
(PDF)

**S2 Appendix. Derivation of the update rules for the models.**
(PDF)

**S3 Appendix. Experiments on a simple nonlinear model.**
(PDF)

**S1 Fig. The impact of the number of inference steps on the velocity and position dimensions in the tracking experiment.** All values are with arbitrary units (a.u.).
(PDF)

**S2 Fig. The performance on the other two dimensions in both hidden and observation levels in the tracking experiment, where *A* and *C* are learnt.** All values are with arbitrary units (a.u.).
(PDF)

## Acknowledgments

The authors thank Sebastian Klavinskis-Whiting for supplying the movie snippets.

## Author Contributions

**Conceptualization:** Beren Millidge, Rafal Bogacz.

**Formal analysis:** Beren Millidge, Mufeng Tang, Mahyar Osanlouy, Nicol S. Harper.

**Methodology:** Nicol S. Harper.

**Resources:** Nicol S. Harper.

**Software:** Beren Millidge, Mufeng Tang, Mahyar Osanlouy.

**Writing – original draft:** Beren Millidge, Mufeng Tang, Mahyar Osanlouy, Rafal Bogacz.

**Writing – review & editing:** Nicol S. Harper.

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
