## [Decision Letter · Decision Letter 0]

8 Jul 2023

Dear Prof. Bogacz,

Thank you very much for submitting your manuscript "Predictive Coding Networks for Temporal Prediction" for consideration at PLOS Computational Biology.

As with all papers reviewed by the journal, your manuscript was reviewed by members of the editorial board and by several independent reviewers. In light of the reviews (below this email), we would like to invite the resubmission of a significantly-revised version that takes into account the reviewers' comments.

We cannot make any decision about publication until we have seen the revised manuscript and your response to the reviewers' comments. Your revised manuscript is also likely to be sent to reviewers for further evaluation.

Sincerely,

Peter E. Latham

Academic Editor

PLOS Computational Biology

Daniele Marinazzo

Section Editor

PLOS Computational Biology

Reviewer's Responses to Questions

**Comments to the Authors:**

Reviewer #1: The paper presents a Kalman filter model of temporal prediction. A theoretical derivation is provided showing how a predictive coding network approximates the Kalman filter, and results are presented demonstrating the ability of the model to learn simple forms of dynamical behavior in low-dimensional systems.

I find this paper extremely interesting, and also inspiring, from a theoretical perspective. The idea is long overdue - as the authors point out, the original predictive coding model of Rao & Ballard, though framed as a Kalman filter, did not actually model temporal structure. The model presented here thus can be seen as finishing that piece of work making it truly predictive in time. Personally I enjoyed reading the paper and learned alot from it.

On the other hand, as a computational biology paper, it leaves something to be desired in connecting in a genuine and compelling way to outstanding questions in neuroscience. For example, how could you map this model onto a population of neurons in V1 (or any sensory system)? could it learn to predict from spatiotemporal patterns in images? Or is there any neural system at all in the brain that you could map this model onto? I believe there is, but the authors say very little or nothing about this. The demonstrations shown in Results are all for extremely low-dimensional systems, for example with three neurons for position, velocity and acceleration. How to map any of these results onto a neural population that would be doing these things - even hypothetically - is not addressed in this paper.

So my concern is one of suitability for the journal as opposed to quality per se: while the paper theoretically is interesting and important, it seems rather contrived from a neuroscience point of view. It would be excellent for a venue such as Neural Computation or NeurIPS.

Specific comments:

There is much made about the disconnection between two different time-scales: the index k that indexes x(k), x(k-1), etc. vs. the time over which optimization is performed. Also the problem of having to "memorize" the past state x_k-1 is discussed. But is this really a problem? In any real neural system, you could have neurons evolving at different timescales. Some are lagged, some non-lagged (as you see in LGN for example). And the sensory input evolves at one timescale, while the rate at which neurons internally can change their activity could evolve at a faster timescale. For example Reinagel has shown that LGN neurons can encode with much higher temporal precision than the timescale of visual signals captured by the cones. The paper would be much more interesting from a neurobiological perspective if it could seamlessly integrate these different timescales into one model, rather than the algorithmic solution of having a "fake time" advancing k-1 to k, and another time for optimization.

The under-determined nature of the solution discussed with respect to Figure 4 seems to point to the fact that the model is not identifiable. Part of this would stem from the assumption of a Gaussian prior. It may thus be desirable to think about using sparse priors over the latent variables x.

Zipser and Williams ref: their original work on this is from 1989:

Williams, R. J., & Zipser, D. (1989). A learning algorithm for continually running fully recurrent neural networks. Neural computation, 1(2), 270-280.

Reviewer #2: **************************************************************

General Summary: ************************

This article develops a neural temporal model based on predictive coding, particularly based on the classical notions of the neural Kalman filter (KF) of Rao 1999, which focuses on inferring and tracking the state of the environment as well as predicting future sensory observations given noisy sensory data inputs. The authors conduct some small-scale synthetic/simulated data experiments to evaluate and understand behavioral/performance aspects of their model.

Strengths: ****************************

+ I enjoyed the starting treatment that walks the reader through some of the foundations/underpinnings of the model, i.e., the HMM and variational free energy treatment in Section 1. I think for readers unfamiliar with PC but familiar with HMMs/KFs and cost functions, this can help a bit in easing them into the framing.

+ I thought the theoretical relationships to the KF were a good touch and useful for the paper. The discussions about the neural implementation were further appreciated (and relevant to this journal overall).

+ The direction/study of how PC framed for temporal forecasting/prediction (building on classical work such as the Rao & Ballard model of 1997) is important and interesting.

Weaknesses: **************************

- Why were the precision matrices, which are arguably quite important, not empirically explored especially on the smaller scale/toy problems examined in this work? I think they should be experimented and studied particularly in this paper (it is certainly understandable that the precision matrices would become cumbersome in higher dimensional problems, but for really small-scale problems I would expect the experimenters/authors to not omit them or, at the very least investigate their influence as precision-weighted error is quite an important element of PC, temporal PC variants, and free energy optimization in general). Given that these are discussed quite a bit throughout, I feel the reader is somewhat (accidentally) misled when the authors simply focus on the simplified temporal PC filters -- and merely point to other works/offload to other static PC models that implement the covariance/precision weighting -- whereas other published efforts (even if these could argued to have their own limitations, in their own ways, as the authors point out early in the text) do attempt to incorporate some form of the precision-weighting (even in the form of scalar precision or diagonal covariance matrix approximations). More importantly, the original PC KF model included these precision weightings as part of the Kalman gain and learned these dynamically (and since the authors' model is largely built on top of the original Rao & Ballard 1997 KF model, which dealt with the precision-weighting, I would expect to see these learned and/or ablated or studied). The authors should conduct experiments (despite some of the theoretical points made in the appendix) in the main paper investigating estimation of the precision (even if using approximations/simplifications of it, such as the recurrent approximation in Tang et al., 2023).

- This manuscript (as well as the appendix) could do with a heavy revision (it feels a bit "raw" in some spots throughout) - it is plagued with quite a few awkward bits of language, instances of casual language (such as use of contractions) which is not appropriate for formal scientific publications, and contains many typos/errors. I have provided an extensive list of error/bad phrasings I caught, but I definitely missed many more and, since it is the job of the authors to thoroughly clean up the language and ensure it is clean, free of error, and flows smoothly, I think this text needs some significant revision before it is ready for publication. (I stopped collecting all the errors/typos/bad phrases I found after a certain point, given the length of the manuscript and leave the rest for the authors to thoroughly and carefully clean up and revise the language to improve flow and reduce error/bad phrasing/formatting issues/inconsistencies). See my but certainly non-exhaustive yet substantial list of errors/mistakes I tried providing corrections/suggestions for.

- While the experimental (synthetic/controlled) simulations for the tracking problems are nice, I do expect to see (or if this is done, made explicit and very clear) in the "Learning" subsection experiments how PC with and without learning its covariance/precision matrices performs (with some possible discussion of issues with learning with precision weighting if there are some that are encountered - I know from experience learning/adapting precision matrices can be tricky in practice and often involve further approximation in order to scale up so it would be useful for the readers to understand the issues even with smaller problem dimensionalities how working with estimates of the precision weighting, such as even using the referenced alterative implementation of Tang et al., 2023 -- I think given this paper's focus, incorporating and studying this precision estimation, if the typical matrix inversion and its approximations prove to be too unstable, it behooves the authors to incorporate some estimation and usage of the precision matrices).

- Given the focus on smaller scale and synthetic tracking problems, I would also expect to see a comparison to a few other relevant estimators, such as scented/extended KF models too (these are the requisite baselines and might, in the worst case they do really, serve as possible goals or upper bounds for readers to consider - since KF/other estimators employ other linear algebraic machinery that would be hard to imagine being implemented in biological neural circuitry). At least including an auto-regressive predictive model (e.g., vector AR/ARIMA) and/or other reasonable time-series estimator/predictor would add some relevant context too (or, if a latent state temporal model is desired, why not include a literal HMM for comparison - it would be good to see how an HMM, which is not learned online typically, would compare and thus highlight the strength of the online-inferred/learned neural KF/PC model).

- How would this approach scale to higher dimensional latent state and observation state problems? While the related efforts referenced(such as the models applied to ML tasks or neuroscience vision tasks) might have their own limitations/issues, they were developed to handle higher-dimensional spaces which are unfortunately faced by engineers and other specialists in general. The authors should, at least in synthetic simulation, experiment with how performance/behavior of the KF and the temporal PC / neural KF estimators change as dimensionality of the observation (and, ideally, the latent) space increases. The brain is generally facing more complex filtering/tracking problems (of much higher dimensions in both latent and observation space, but at the very least in observation space) and it would be important for this work to least offer some investigation in this direction (as this is an important constraint/limitation of many models that work so well in low-dimensional toy tasks but break down in high-dim problems -- this also relates to the limitations I brought up earlier w.r.t. precision-weighting as precision matrices become quite problematic in very dimensional estimation problems)

- This study should also include at least one (if not two) real-world time series or dynamics dataset to compare the temporal PC model (as well as the Rao & Ballard 1997 neural KF model and 1-2 other KF or time-series estimation models) on as currently the study only conducts simulation studies, which are a nice starting point but not clearly enough for a substantial publication (even in statistics, methodology that is developed often starts with a simulation study and then moves on to a real-world dataset or multiple datasets; even the original Rao & Ballard 1997 neural KF model investigated videos of real-world images, even if small and limited in nature). The authors should at minimum conduct experiments on (with comparisons to the relevant baselines mentioned in this review) real-world temporal data / sequences.

These can be low-dimensional but should not be synthetic simulations as this does not really get at the performance (with respect to both benefit and limitation) of temporal PC and temporal models in generally. To really strengthen this work, given the PC is a good candidate for modeling biological neural processing, it'd be ideal to evaluate its performance on at least one instance of raw unstructured temporal data (again, even if the image space dimensionality is controlled through dimensionality reduction or scaling/sampling), e.g., pixels in video sequences much as has been even in the classical models such as Rao & Ballard 1997 (neural KF PC model) where computing resources were much, much more constrained compared to today.

**** Typos/Revisions Found (though others might have been missed): *****

Abstract:

"problems the brain faces" should be "problems that the brain faces"

changing stimuli, and it is not yet" should be something akin to: "changing stimuli. However, it is not yet"

Author Summary:

Change "has often been relatively neglected" to "has often been neglected"

Page 2:

"that learning in predictive coding model requires" should be "that learning in predictive coding models require" (plurality grammatical error)

Page 4:

"individual images with much correlation" should be something akin to "individual images containing a great deal of correlation"

"process them online – i.e. as the inputs" should be "process them online, i.e. as the inputs" (no dash, since there is a supporting comma-delineated phrase)

(page 4/5) "sequentially, as biological systems need to do." should be something akin to "sequentially as biological systems appear to do."

Page 5:

"have proposed the notion of extending" should be "proposed the notion of extending"

"derivatives into the" should be "derivatives in the"

"specially designed network" should be "specialized network"

Page 6:

"contains much mathematical notation" should "contains a great deal of mathematical notation and symbols"

Page 7:

Figure 1 caption - "assumed by the temporal predictive coding networks" should be "assumed by a temporal

predictive coding network"

Page 8:

"which we don't" change to "which we do not" (do not use casual language in formal publications, and contractions are generally used in casual English speech/language)

Page 9:

"can be considered as" should either be "can be considered to be" OR "can be treated as"

At end of Equation 4, there is missing a period (generally mathematical equations flow as if part of the sentence structure, so if no English language is to follow it in the text below, such as to further define operators or symbols or to add intuition, generally put a period after it). Other examples include 6, 7 and many others later on, check other equations to ensure this type of formatting/formality.

Page 10:

"assumptions in" should be either "assumptions made in" or "assumptions underlying"

Section title "Algorithm in the model" is awkward and does not read right - try something such as "Algorithm for model inference and learning"

Page 11:

"to static predictive coding" should be "to that of predictive coding"

Page 12:

"for a single step" should be "in a single step"

"Given this \\hat{x}_k we can update the" should be "Given this \\hat{x}_k, we can then update the"

"rules and dynamics we have" should be "rules and dynamics that we have"

"In this section we present" should be "In this section, we present" (please check to fix errors related to not closing certain types of appositive or introductory phrases that generally use commas)

"exactly, and" change to "exactly and" (no comma used here)

Page 13:

"rules we have" to "rules that we have"

"level, and cause" to "level and cause"

Page 14:

Error in Figure 2 - apical dendrites according to the visual are ellipses (including your key) but the caption says "rectangles" - correct your caption and figure and ensure they are unified and clear to avoid confusing the reader

Page 15:

"we update state" must be "we update the state"

Page 16:

"where the observation are" should be "where the observations are"

Page 17:

"the optimum estimated hidden" should be "the optimal estimated hidden"

"to achieve the optimum" would read better as "to reach the optimum"

"that in practice on" to "that, in practice, on"

Page 18:

"we demonstrate that despite" to "we demonstrate that, idespite"

Page 19:

"new information we have" to "new information that we have"

Page 20:

"temporal predictive coding model i.e.," change to "temporal predictive coding model, i.e.,

Page 22:

"with those the brain must" to "with those that the brain must"

Page 27:

"data well, and we wished to test if it can do" to "data well and we examined whether or not it would be able to do"

Page 28:

"Both a liner and" change to "Both a linear and"

"a simulation data" to "a simulated data generating process"

Page 30:

"plotted the mean learnt matrix" to "plotted the mean of the learnt matrix"

Page 31:

"were initialized as identity matrix and zero," to "were initialized to the identity matrix and the zero matrix,"

"The learning was performed as in the" to "The learning was performed in the same fashion as in the"

Page 36:

"compare if learning rate" change to "compare if the learning rate"

Page 39:

"we have identified representation of subjective uncertainty as the key difference between the Kalman filter and the temporal predictive coding networks" should be corrected to: "we have identified that the representation of subjective uncertainty is the key difference between the Kalman filter and temporal predictive coding networks"

Reviewer #3: The review has been uploaded as an attachment.

**Have the authors made all data and (if applicable) computational code underlying the findings in their manuscript fully available?**

Reviewer #1: None

Reviewer #2: None

Reviewer #3: Yes

PLOS authors have the option to publish the peer review history of their article (what does this mean?). If published, this will include your full peer review and any attached files.

Reviewer #1: No

Reviewer #2: No

Reviewer #3: No
---

## [Decision Letter · Decision Letter 1]

12 Mar 2024

Dear Prof. Bogacz,

We are pleased to inform you that your manuscript 'Predictive Coding Networks for Temporal Prediction' has been provisionally accepted for publication in PLOS Computational Biology.

Best regards,

Peter E. Latham

Academic Editor

PLOS Computational Biology

Daniele Marinazzo

Section Editor

PLOS Computational Biology

Reviewer's Responses to Questions

**Comments to the Authors:**

Reviewer #1: The paper has been revised and now includes a demonstration of how dynamic visual receptive fields in V1 could be learned from this framework. These results are interesting and help to move the paper toward being more biologically relevant. I still think there is more that could be done to tie to neuroscience though - for example, what does the A matrix learn for natural movies? This might provide an interesting prediction about the role of horizontal connections in V1. Also, there is other work that should be cited along these lines:

Pachitariu, M., & Sahani, M. (2012). Learning visual motion in recurrent neural networks. Advances in Neural Information Processing Systems, 25.

Pachitariu, M., & Sahani, M. (2017). Visual motion computation in recurrent neural networks. bioRxiv, 099101.

Also a comment about the learned receptive fields shown in panel B of Figure 6: they are rather odd looking in that the excitatory and inhibitory regions are so elongated and highly parallel, almost cartoon-like. One does not normally see this from sparse coding on natural images (which this algorithm essentially is, along with the dynamics provided by the A matrix). At first I was puzzled by this, but then noticed there is an L1 penalty on the C matrix. That perhaps explains why they are so tightly localized and perfect looking. I think it would be worth noting this in the text as I imagine other readers familiar with learning on natural images will be puzzled by this too.

Finally, I note that the reference given for Williams & Zipser is incomplete:

Williams RJ, Zipser D. Gradient-based learning algorithms for recurrent. Backpropagation: Theory, architectures, and applications. 1995;433:17.

what is the journal? and why not cite their original paper on this:

Williams, R. J., & Zipser, D. (1989). A learning algorithm for continually running fully recurrent neural networks. Neural computation, 1(2), 270-280.

Reviewer #2: I am happy with the authors' general set of revisions (including the responses to the other reviewers) and the general addressal of my request for the additional experiments and further examination of the precision matrices. In particular, I appreciated the additional investigation of how the model performed on the natural image video samples (that originated from the time of Bruno's classic work!) for the more complex experiment I wanted to see.

Thank you for your response and updates to the manuscript.

For reproducibility purposes, please make the data (or viable links to their sources), particularly data such as natural image videos, available on your github repo -- when I went to the github link to poke around I found the data loading routines/scripts for data such as the natural images but ( could not apparently find the natural image videos themselves (I might have missed these, possibly). Please either place your used data points in a /data/ folder if it's small enough or point to the original natural image video download source with a link on the github README otherwise.

** Additional small note: I still personally think the comparison to KF models like the extended/scented variants would have been important and useful to include in this paper (particularly on the smaller dimensional problems) given how important these classical models are. Hopefully, the authors will conduct such an investigation/comparison in future related efforts (possibly mentioning this in their conclusion/outlook).

Reviewer #3: All my comments have been addressed appropriately.

**Have the authors made all data and (if applicable) computational code underlying the findings in their manuscript fully available?**

Reviewer #1: Yes

Reviewer #2: None

Reviewer #3: Yes

PLOS authors have the option to publish the peer review history of their article (what does this mean?). If published, this will include your full peer review and any attached files.

Reviewer #1: **Yes: **Bruno Olshausen

Reviewer #2: No

Reviewer #3: No

---

## [Editor Report · Acceptance letter]

25 Mar 2024

PCOMPBIOL-D-23-00771R1 

Predictive Coding Networks for Temporal Prediction

Dear Dr Bogacz,

I am pleased to inform you that your manuscript has been formally accepted for publication in PLOS Computational Biology. Your manuscript is now with our production department and you will be notified of the publication date in due course.

With kind regards,

Olena Szabo
